# DISTILLING GANS WITH STYLE-MIXED TRIPLETS FOR X2I TRANSLATION WITH LIMITED DATA

**Yaxing Wang[1,2], Joost van de Weijer[2], Lu Yu[3],*, Shangling Jui[4]**

[1] College of Computer Science, Nankai University, China
[2] Computer Vision Center, Universitat Autònoma de Barcelona, Spain
[3] School of Computer Science and Engineering, Tianjin University of Technology, China
[4] Huawei Kirin Solution, China

{yaxing,joost,luyu}@cvc.uab.es, jui.shangling@huawei.com

## ABSTRACT

Conditional image synthesis is an integral part of many X2I translation systems, including image-to-image, text-to-image and audio-to-image translation systems. Training these large systems generally requires huge amounts of training data. Therefore, we investigate knowledge distillation to transfer knowledge from a high-quality unconditioned generative model (e.g., StyleGAN) to a conditioned synthetic image generation modules in a variety of systems. To initialize the conditional and reference branch (from a unconditional GAN) we exploit the style mixing characteristics of high-quality GANs to generate an infinite supply of style-mixed triplets to perform the knowledge distillation. Extensive experimental results in a number of image generation tasks (i.e., image-to-image, semantic segmentation-to-image, text-to-image and audio-to-image) demonstrate qualitatively and quantitatively that our method successfully transfers knowledge to the synthetic image generation modules, resulting in more realistic images than previous methods as confirmed by a significant drop in the FID. Code is available in https://github.com/yaxingwang/KDIT.

## 1 INTRODUCTION

Conditional image synthesis, also *X2I translation*, maps from an input domain (e.g. text, audio, segmentation maps, etc.) to the image domain. Benefiting from GANs (Goodfellow et al., 2014) and its follow-up improved versions (Gulrajani et al., 2017; Kang & Park, 2020; Salimans et al., 2016), they obtain remarkable performance on a wide variety of image synthesis tasks: image to image (I2I) (Lee et al., 2018; Zhu et al., 2017), audio to image (Chen et al., 2017; Wang et al., 2020a), text to image (Hu et al., 2021; Li et al., 2019; 2020; Radford et al., 2021; Zhang et al., 2017a) and semantic segmentation map to image (Isola et al., 2017; Wang et al., 2018). Despite impressive leaps forward for a variety of image synthesis tasks, there are still important challenges. Specifically, to obtain good results, existing works rely on large labelled datasets. Labeling these datasets is both laborious and time-consuming, considerably reducing the practical impact of these methods. It is noteworthy to observe that many of these models (Zhang et al., 2017a) apply transfer learning to the text and audio encoders (e.g. using a pretrained LSTM (Reed et al., 2016) model for text and pretrained GRU (Merkx et al., 2019) model for audio), however they train the image synthesis decoder from scratch. This happens because there are no established methods to transfer pretrained GANs to conditional image decoders; an omission which we aim to address in this paper.

In this paper, we investigate knowledge transfer for a variety of conditional image synthesis tasks. Traditional knowledge transfer for conditional image synthesis is often not possible, because there might not be a pretrained network available for the desired translation task (e.g. at the moment no high-quality pretrained network for segmentation map-to-image translation is available). It would therefore be preferable if the wide variety of high-quality GANs available for image generation could be exploited for X2I. Recent works (Wang et al., 2021; 2020b) leveraged a pretrained GAN to initialize an I2I translation model, managing to transfer knowledge to different image synthesis

---

*The corresponding author.

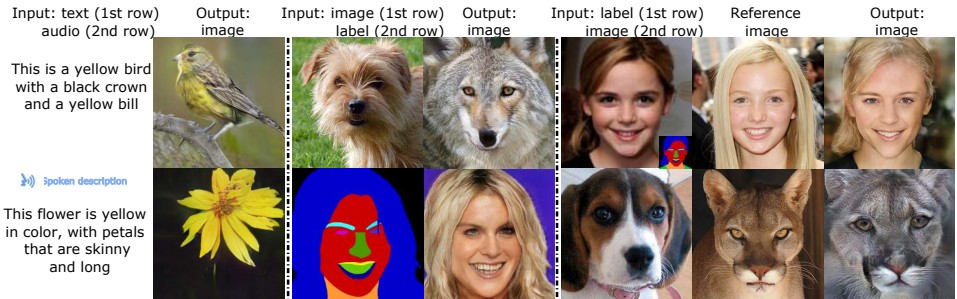

Figure 1: Various examples of our method. Using the proposed knowledge transfer method, we are able to generate high-quality images for several image synthesis tasks.

tasks. These methods, however, suffer from three problems: (I) They can only be used for I2I translation and do not generalize to other conditional image synthesis tasks. (II) They are GAN architecture-specific approaches, requiring the GAN architecture within the X2I system to be exactly the same as that of the pretrained GAN. This limits transfer to current state-of-the-art X2I systems for which no similar GAN architecture exists (like for example StarGANv2 (Choi et al., 2020) for I2I). (III) X2I systems are based on a conditional GAN, however existing methods for knowledge transfer for I2I do not initialize the conditional branch during the transfer, and therefore this has to be learned from scratch during the finetuning on the target dataset. Learning this on small target datasets can be problematic.

To address the aforementioned problems, we propose several improvements for knowledge transfer to X2I systems: (I) we are the first to investigate knowledge transfer for X2I translation. Therefore, we propose a novel, unified transfer learning method, which can be used for varying kinds of conditional image synthesis tasks (Figure 1) which is based on generated images and therefore does not require any real data. (II) The student generator does not need to have the same architecture as the pretrained GAN. Therefore, we can use well-devised specific image synthesis architectures (e.g., SPADE (Park et al., 2019) and StarGANv2 (Choi et al., 2020)) by distilling knowledge from the pretrained teacher GAN (e.g., StyleGAN) to the task-specific student GAN. (III) We use the *style mixing* characteristic of StyleGAN to create *style-mixed triplet* data, which are used to transfer the knowledge efficiently to both I2I and X2I translation models. Furthermore, we propose a semantic diversity loss based on the style-mixed triplet, which contributes to learn the semantic information of the output image.

We perform experiments on a wide variety of image synthesis tasks, including text-to-image, audio-to-image, segmentation map-to-image and I2I translations. We demonstrate the efficiency of the proposed knowledge distillation method, providing qualitative and quantitative results. We prove that the single pretrained GAN model can be universally used in varying specific task model. Additionally, leveraging the *style mixing* character of StyleGAN, further improves I2I translation performance.

## 2 RELATED WORK

**GAN-based Conditional Image Synthesis.** Benefiting from the advances in GANs and its variants in recent years, conditional image synthesis (also called X2I translation) research has developed rapidly. Two typical approaches have been investigated for GAN-based image synthesis, namely, unsupervised (Kim et al., 2017; Yi et al., 2017) and supervised image generation (Park et al., 2019; Zhang et al., 2017b; Zhou et al., 2020). The latter inputs conditional information (e.g. text, audio, image, segmentation map etc.) to synthesize images which contain the corresponding semantic information (i.e. the conditional information). Specifically, text-to-image translation (Hu et al., 2021; Li et al., 2020; Zhang et al., 2017a) aims to synthesize high-realistic images which are semantically consistent with the text descriptions. Recent work (Hu et al., 2021) introduces semantic-spatial batch normalization to better exploit the text information. Similar to text-to-image translation, both audio-to-image translation (Chen et al., 2017; Wang et al., 2020a) and segmentation map-to-image translation (Bau et al., 2020; Park et al., 2019) aim to learn a mapping from the audio/segmentation map to the output image. Different to the above image synthesis tasks, image-to-image transla-

tion (Park et al., 2020; Zhu et al., 2017) performs projection from the source to the target image domain. In this paper, we explore transfer learning from GANs to a variation of *conditional* image synthesis tasks.

**Transfer learning.** A considerable research effort has investigated transferring knowledge for both discriminative (Donahue et al., 2014; Hinton et al., 2014; Xie et al., 2015; Yu et al., 2019a; Zhou et al., 2022; Zhao et al., 2020a) and generative tasks (Noguchi & Harada, 2019; Zhao et al., 2020b). There also exist several approaches (Goetschalckx et al., 2019; Jahanian et al., 2020) which focus on the image manipulation based on the pretrained GAN. Given a target semantic attribute they aim to manipulate the output image of a pretrained GAN. However, these methods do not focus on transfer learning for target data. Some other methods (Abdal et al., 2019; Zhu et al., 2020a) reverse the given image into the input latent space of the pretrained GAN (e.g., StyleGAN), and manage to restructure the target image by optimization of the latent representation. Recent work (Shocher et al., 2020; Wang et al., 2021; 2020b) performed knowledge transfer from a pretrained classification model (e.g., VGG (Simonyan & Zisserman, 2014)) or the discriminator (BigGAN (Brock et al., 2019)) for I2I translation. However, both DeepI2I (Wang et al., 2020b) and TransferI2I (Wang et al., 2021) require that the GAN architecture is identical with the generator used in the I2I architecture. As a consequence, these methods cannot be applied to well-designed I2I translation architectures (like starGANv2 (Choi et al., 2020)) since they do not use a standard GAN architecture. The proposed method could address these problems.

## 3 KNOWLEDGE TRANSFER FOR X2I

**Problem setting.** Our goal is to transfer knowledge from a pretrained high-quality unconditional GAN to an X2I translation system for the case when training data is limited. The proposed method consist of two stages. In the first stage we perform the data-free knowledge transfer method explained in Sec. 3.2 and 3.3. This stage does not require any target data. In the second stage, we apply a standard finetuning of this distilled model on the target dataset. Distillation can be used to perform knowledge transfer to other architectures, however, there is little target data available. Fortunately, the pretrained GAN can generate infinite data for data distillation, meaning that we do not require access to any real data (i.e., *data-free*). Also, we transfer knowledge from an unconditional GAN (e.g., StyleGAN) to a conditional GAN (e.g., I2I translation); requiring us to propose techniques to initialize the conditional branch. Exploiting the style-mixed characteristic (Sec. 3.1) of StyleGAN we propose two solutions: one for I2I (Sec. 3.2) and another one for X2I (Sec. 3.3). Our method consists of two stages:(1) *transfer leaning* without any real data (i.e., style-mixed triplets in Sec. 3.1, I2I translation in Sec. 3.2 and X2I translation in Sec. 3.3 ) and (2) *finetunning* with the real target data.

### 3.1 STYLE-MIXED TRIPLETS

Benefiting from the *style mixing* ((Figure 2 (a))) characteristic of StyleGAN, we are able to create an infinite amount of *triplet* images from any of the two domains. Let $\mathbf{z_t} \in \mathbb{R}^{\mathbf{Z}}$ indicates the input noise of the pretrained generator $G_T$ (teacher). given the input noises $\mathbf{z_t^1}$ and $\mathbf{z_t^2}$, the *mapping network* $M_T$ of the StyleGAN generator encodes them to a style vector $\mathbf{s_t^1} = M_T(\mathbf{z_t^1})$ and $\mathbf{s_t^2} = M_T(\mathbf{z_t^2})$. We define $G_T(\mathbf{z}) = G_T'(M_T(\mathbf{z}))$ where $G_T'$ takes styles vectors as an input to each of its layers. We further feed these style vectors to the teacher generator $G_T'$ to output $\mathbf{x_t^1} = G_T'(\mathbf{s_t^1})$ and $\mathbf{x_t^2} = G_T'(\mathbf{s_t^2})$ (where $\mathbf{x_t^1}, \mathbf{x_t^2} \in \mathcal{X}$ and image domain $\mathcal{X} = \mathbb{R}^{H \times W \times 3}$) respectively. Here the function $\Phi(\mathbf{s_t^1}, \mathbf{s_t^2})$ selects to forward $\mathbf{s_t^1}$ to the layers of $G_T'$ important for the content of the generated image (the bottom layers), and $\mathbf{s_t^2}$ to those important for the style (the top layers)[1]. Interestingly, a new image $\mathbf{y_t} = G_T'(\Phi(\mathbf{s_t^1}, \mathbf{s_t^2}))$ (where $\mathbf{y_t} \in \mathcal{Y}$ and image domain $\mathcal{Y} = \mathbb{R}^{H \times W \times 3}$) contains mixed information with respect to content and style of both inputs $\mathbf{x_t^1}$ and $\mathbf{x_t^2}$. Based on this observation above, we propose to leverage these style-mixed triplets $(\mathbf{x_t^1}, \mathbf{x_t^2}, \mathbf{y_t})$ to perform distillation for the I2I and X2I translation models without the need of any real triplet training data. [2]

---

[1]We use the the style vector $\mathbf{s_t^1}$ in the first four layers of the generator, and do $\mathbf{s_t^2}$ in the following layers.

[2]In practice, during experiments we generate style-mixed triplets online in the minibatch.

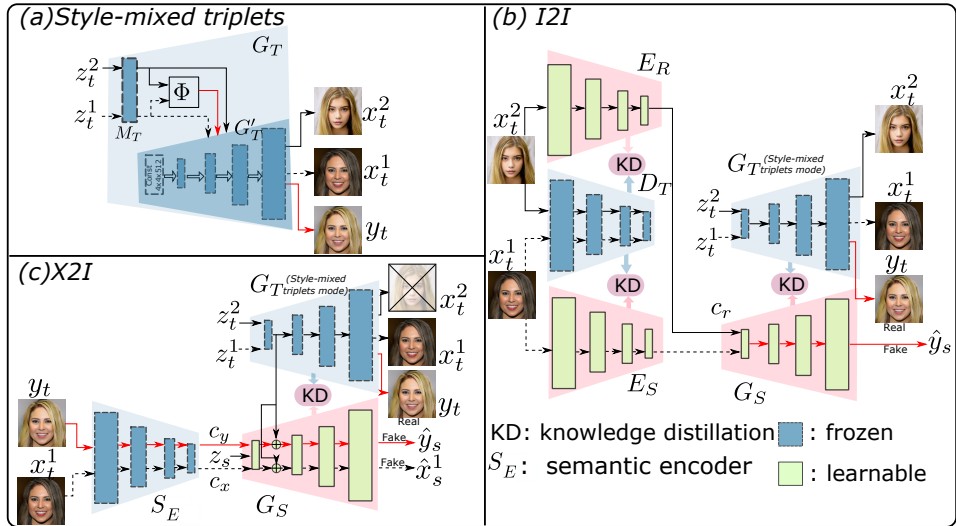

Figure 2: (a) Synthesis of style-mixed triplets ($\mathbf{x_t^1}$,$\mathbf{x_t^2}$,$\mathbf{y_t}$). (b) Overview of our data-free knowledge distillation for I2I translation. We transfer knowledge from the teacher pretrained discriminator $D_T$ to both the student encoder $E_S$ and the reference encoder $E_R$. In addition, we transfer from the pretrained generator $G_T$ to the student generator $G_S$. Here $\mathbf{x_t^1}$ is the input image, $\mathbf{x_t^2}$ is the reference image, and $\mathbf{y_t}$ is the target image. (c) Overview of our data-free distillation for X2I translation. The semantic encoder $S_E$ aims to extract the attribute embedding, and further control the semantic characteristic of the student generator output. The semantic encoder is identical to the teacher discriminator and initialized by it.

## 3.2 DATA-FREE KNOWLEDGE DISTILLATION FOR I2I TRANSLATION

We aim to leverage the style-mixed triplets to perform knowledge transfer for the I2I model. Let $\mathbf{x_t} \in \mathcal{X}$ and $\mathbf{y_t} \in \mathcal{Y}$ indicate the two domains. We expect to map an image $\mathbf{x_t}$ from domain $\mathcal{X}$ into another sample $\mathbf{y_t}$ which mimics to $\mathcal{Y}$. Note we never get access to any real data. Figure 2 (b) shows the knowledge distillation framework for I2I translation. Our framework is composed of six neural networks: a pretrained teacher GAN (consisting of $G_T$ and $D_T$), and an I2I model with a student generator $G_S$, a student discriminator $D_S$, a student encoder $E_S$ that extracts the content information from the input image $\mathbf{x_t^1}$, and a reference encoder $E_R$ that extracts the style information from the reference image $\mathbf{x_t^2}$. Similarly to StarGANv2, we investigate two cases of I2I translation: the style representation is from noise (i.e., *latent-guided synthesis*) and from the reference image (i.e., *reference-guided synthesis*). Here we mainly focus on *reference-guided synthesis*, since *latent-guided synthesis* can be adapted easily by replacing the reference encoder with several fully connection layers, which take the noise as input.

Both the teacher discriminator $D_T$ and the student encoder $E_S$ take the input image $\mathbf{x_t^1}$ as input, extracting the hierarchical representation $\{D_T(\mathbf{x_t^1})_l\}$ and $\{E_S(\mathbf{x_t^1})_l\}$. Here $D_T(\mathbf{x_t^1})_l$ represents the output of the $l_{th}$ ResBlock [3]. We align them via the feature-based knowledge distillation loss:

$$\mathcal{L}_{kdl}^e = \sum_l \gamma_l \left\| D_T(\mathbf{x_t^1})_l - E_S(\mathbf{x_t^1})_l \right\|_1 \tag{1}$$

where parameters $\gamma_l$ are scalars to balance the terms. We set them to 0.1. When the teacher and student dimensions are not the same, an additional layer can be introduced to map the teacher output to the desired dimensions (similar as the hint distillation used in (Romero et al., 2015)).

Next, we take the image $\mathbf{x_t^2}$ for both the pretrained discriminator $D_T$ and the reference encoder $E_R$, extracting the hierarchical representation $\{D_T(\mathbf{x_t^2})_l\}$ and $\{E_R(\mathbf{x_t^2})_l\}$, both of which we encourage to be aligned via the feature-based knowledge distillation loss

$$\mathcal{L}_{kdl}^r = \sum_l \tau_l \left\| D_T(\mathbf{x_t^2})_l - E_R(\mathbf{x_t^2})_l \right\|_1 \tag{2}$$

---

[3] After each ResBlock the feature resolution is half of the previous one in both encoder and discriminator, and two times in generator.

where parameters $\tau_l$ are scalars which balance the terms. We set them to 0.1.

Finally, taking the output of the student encoder $E_S(\mathbf{x_t^1})$ and the output of the reference encoder $E_R(\mathbf{x_t^2})$, for the student generator $G_S$ we obtain the hierarchical representation $\{G_S(E_S(\mathbf{x_t^1}), E_R(\mathbf{x_t^2}))_l\}$ and the output image $\hat{\mathbf{y}}_\mathbf{s} = G_S(E_S(\mathbf{x_t^1}), E_R(\mathbf{x_t^2}))$, both of which we encourage to align with the hierarchical representation $\{G'_T(\Phi(\mathbf{s_t^1}, \mathbf{s_t^2}))_l\}$. We also align the output image $\mathbf{y_t} = G'_T(\Phi(\mathbf{s_t^1}, \mathbf{s_t^2}))$ with $\mathbf{y}_t$:

$$\mathcal{L}_{kdl}^g = \sum_l \alpha_l \left\| G'_T(\Phi(\mathbf{s_t^1}, \mathbf{s_t^2}))_l - G_S(E_S(\mathbf{x_t^1}), E_R(\mathbf{x_t^2}))_l \right\|_1 + \beta \left\| \mathbf{y}_t - \hat{\mathbf{y}}_\mathbf{s} \right\|_1 . \tag{3}$$

where parameters $\alpha_l$ and $\beta$ are scalars to balance the terms. We set them to 0.1.

We also employ the following GAN loss (Goodfellow et al., 2014) to optimize this problem:

$$\mathcal{L}_{adv} = \mathbb{E}_{\mathbf{y_t} \sim \mathcal{Y}} \left[ \log D_S \left( \mathbf{y_t} \right) \right] + \mathbb{E}_{\hat{\mathbf{y}}_\mathbf{s} \sim \mathcal{Y}} \left[ \log(1 - D_S \left( \hat{\mathbf{y}}_\mathbf{s} \right) \right], \tag{4}$$

The full objective function of our model is:

$$\min_{E_S, G_S} \max_{D_S} \lambda_{adv} \mathcal{L}_{adv} + \lambda_{kdl}(\mathcal{L}_{kdl}^e + \mathcal{L}_{kdl}^r + \mathcal{L}_{kdl}^g) \tag{5}$$

where $\lambda_{adv}$ and $\lambda_{kdl}$ are hyper-parameters to balance their relative importance. We set them to 1.

In conclusion, to improve knowledge transfer for I2I, we have presented two novel contributions. Firstly, we have shown that the pretrained StyleGAN can be used to generate an unlimited number of style-mixed triplets which we can use to perform the training. We are the first to exploit the style-mixing possibilities of StyleGAN to transfer knowledge to I2I systems. Secondly, we have proposed a distillation approach to transfer knowledge from the GAN to the I2I model. We name this data-free distillation because no real data is required and it is based on generated data. our method can transfer knowledge between generators with different architectures.

### 3.3 Data-free knowledge distillation for X2I translation

Here we show how the proposed data-free knowledge distillation can be generalized to X2I translation problems. We will perform the distillation based on a noise input to the student generator. For X2I translation, the conditional information $\mathbf{c}$ determines the structural information and the semantic information. For instance, in text2image translation on the birds dataset, we take as an input text descriptor (information $\mathbf{c}$), and expect to generate an image which mimics the shape and the appearance of a real bird. However, during the knowledge transfer, the conditional information $\mathbf{c}$ is not available. Therefore, we propose to use the teacher GAN discriminator to extract a semantic encoder of the image that can be used to replace the conditional information $\mathbf{c}$. However, when directly applying this for knowledge distillation, we found that this information can still be ignored by the network (see e.g. Figure 11). We therefore introduce two additional techniques. We again use the style-mixed triplets introduced in the previous section to perform the knowledge transfer and we propose a novel semantic diversity loss to diversify the semantic information of the output when varying the condition $\mathbf{c}$.

As shown in Figure 2(c), we additionally introduce a semantic encoder $S_E$ to extract a pseudo-condition vector $\mathbf{c}$, which assists in distilling the knowledge from the unconditional GAN $G_T$ to the conditional GAN $G_S$. The semantic encoder $S_E$ is identical to the teacher discriminator after removing the last fully connection layer, and initialized by the teacher discriminator, which is well-optimized on the input images $\mathbf{y_t}$ and $\mathbf{x_t^1}$. Utilizing the style-mixed triplets (Sec. 3.1), we extract teacher output $\mathbf{x_t^1} = G'_T(\mathbf{s_t^1})$ and $\mathbf{y_t} = G'_T(\Phi(\mathbf{s_t^1}, \mathbf{s_t^2}))$, and the corresponding hierarchical representation $\{G'_T(\mathbf{s_t^1})_l\}$ and $\{G'_T(\Phi(\mathbf{s_t^1}, \mathbf{s_t^2}))_l\}$, respectively. Note that $G'_T(\mathbf{s_t^1})_1$ and $G'_T(\Phi(\mathbf{s_t^1}, \mathbf{s_t^2}))_1$ are identical, since $\mathbf{x_t^1}$ provides the structural information of $\mathbf{y_t}$. Semantic encoder $S_E$ extracts the semantic attributes from the input images $\mathbf{x_t^1}$ and $\mathbf{y_t}$ respectively, termed as $\mathbf{c_x} = S_E(\mathbf{x_t^1})$ and $\mathbf{c_y} = S_E(\mathbf{y_t})$ respectively.

For the student generator $G_S$, we take as input the teacher's latent representation at the first layer, the output the semantic encoder and the noise. Note, the teacher's latent representation [4] is summed

---

[4]We ignore the latent representation in the second stage (i.e., fintunning on target domain), which means that the student generator has two inputs: the noise and the condition $\mathbf{c}$.

with the corresponding feature of the student generator. When the input of the semantic encoder $S_E$ is $\mathbf{x_t^1}$, we generate the hierarchical representation $\{G_S(\mathbf{z_s}, G'_T(\mathbf{s_t^1})_1, \mathbf{c_x})_l\}$ and final student output $\hat{\mathbf{x}}_\mathbf{s}^1 = G_S(\mathbf{z_s}, G'_T(\mathbf{s_t^1})_1, \mathbf{c_x})$. Similarly, when $\mathbf{y_t}$ is the input of semantic encoder $S_E$, we have the hierarchical representation $\{G_S(\mathbf{z_s}, G'_T(\Phi(\mathbf{s_t^1}, \mathbf{s_t^2}))_1, \mathbf{c_y})_l\}$ and final student output $\hat{\mathbf{y}}_\mathbf{s} = G_S(\mathbf{z_s}, G'_T(\Phi(\mathbf{s_t^1}, \mathbf{s_t^2}))_1, \mathbf{c_y})$.

By conditioning the student generator with the teacher representation, the student generator can output a similar image (this would otherwise be impossible). The loss is defined as:

$$\mathcal{L}_{kdl} = \sum_l \alpha_l \left\| G'_T(\mathbf{s_t^1})_l - G_S(\mathbf{z_s}, G'_T(\mathbf{s_t^1})_1, \mathbf{c_x})_l \right\|_1 + \beta \left\| \mathbf{x_t^1} - \hat{\mathbf{x}}_\mathbf{s}^1 \right\|_1 +$$
$$\sum_l \alpha_l \left\| G'_T(\Phi(\mathbf{s_t^1}, \mathbf{s_t^2}))_l - G_S(\mathbf{z_s}, G'_T(\Phi(\mathbf{s_t^1}, \mathbf{s_t^2}))_1, \mathbf{c_y})_l \right\|_1 + \beta \left\| \mathbf{y_t} - \hat{\mathbf{y}}_\mathbf{s} \right\|_1 \tag{6}$$

where parameters $\alpha_l$ and $\beta$ are scalars to balance the terms.

Yang et al. (2019) proposed a loss to promote diversity of the generated images with respect to changes of the input noise $\mathbf{z}$. Inspired by this work, to address the lack of diversity with respect to the conditional vector, we propose the semantic diversity loss is:

$$\begin{aligned} \mathcal{L}_{srl} &= -\|\hat{\mathbf{x}}_\mathbf{s}^1 - \hat{\mathbf{y}}_\mathbf{s}\|_1 \\ &= -\|G_S(\mathbf{z_s}, G'_T(\mathbf{s_t^1})_1, \mathbf{c_x}) - G_S(\mathbf{z_s}, G'_T(\Phi(\mathbf{s_t^1}, \mathbf{s_t^2}))_1, \mathbf{c_y})\|_1 \\ &= -\|G_S(\mathbf{z_s}, G'_T(\mathbf{s_t^1})_1, \mathbf{c_x}) - G_S(\mathbf{z_s}, G'_T(\mathbf{s_t^1})_1, \mathbf{c_y})\|_1 \end{aligned} \tag{7}$$

Other than (Yang et al., 2019), our loss promotes diversity with respect to changes of the conditional vectors (i.e., $\mathbf{c_x}$ and $\mathbf{c_y}$) while we retain the input noise vector $\mathbf{z_s}$. Since the conditional vector influences the semantic information (e.g., gender, style), we call our loss the semantic diversity loss. The full objective function of our model is:

$$\min_{G_S} \max_{D_S} \lambda_{adv} \mathcal{L}_{adv} + \lambda_{kdl} \mathcal{L}_{kdl} + \lambda_{srl} \mathcal{L}_{srl} \tag{8}$$

where $\lambda_{adv}$, $\lambda_{kdl}$ and $\lambda_{srl}$ are trade-off parameters, and $\mathcal{L}_{adv}$ is defined in Eq. 4.

The proposed initialization procedure is not applicable to the case when the conditioning is a one-hot spatial map (we found the semantic encoder was not able to predict these). Therefore, for this case, we do not initialize the conditional branch and we set $\mathbf{c}$ are $\mathbf{1}$ in the student generator and $\mathbf{0}$ in the student discriminator during the distillation. We found that even for these cases the knowledge transfer led to large performance gains (Table 2).

## 4 EXPERIMENTS

In Sec. 4.1, we investigate knowledge transfer for I2I translation. We explore how the proposed method affects image synthesis by using conditional information $\mathbf{c}$ (i.e., audio, text and semantic segmentation) in Sec. 4.2. For the experimental evaluation, we apply our proposed knowledge transfer method explained in the previous section (i.e the first stage) and use this distilled model for the downstream task where we finetuned it on the target data (i.e. the second stage). We use the pretrained StyleGAN optimized on HHFQ human face (Karras et al., 2019) except for semantic segmentation-to-image(SS2image), which is on AFHQ dataset (Choi et al., 2020). More detailed information and visualization results can be found in the Appendix A and C.

### 4.1 DATA-FREE KNOWLEDGE DISTILLATION FOR I2I TRANSLATION

We first evaluate our method for I2I translation with a reference image, named *reference-guided I2I translation*. Then we verify the proposed method without reference image, called *latent-guided I2I translation*, in which we use random noise to control the style of the output image. We take state-of-the-art StarGANv2 as the student network. Except for the well known metrics: FID (Heusel et al., 2017) and KID (Bińkowski et al., 2018), we also train a real (*RC*) and a fake classifier (*FC*) (Shmelkov et al., 2018) to evaluate the ability to generate class-specific images. RC is trained on real data and evaluated on the generated data and vice versa for FC.

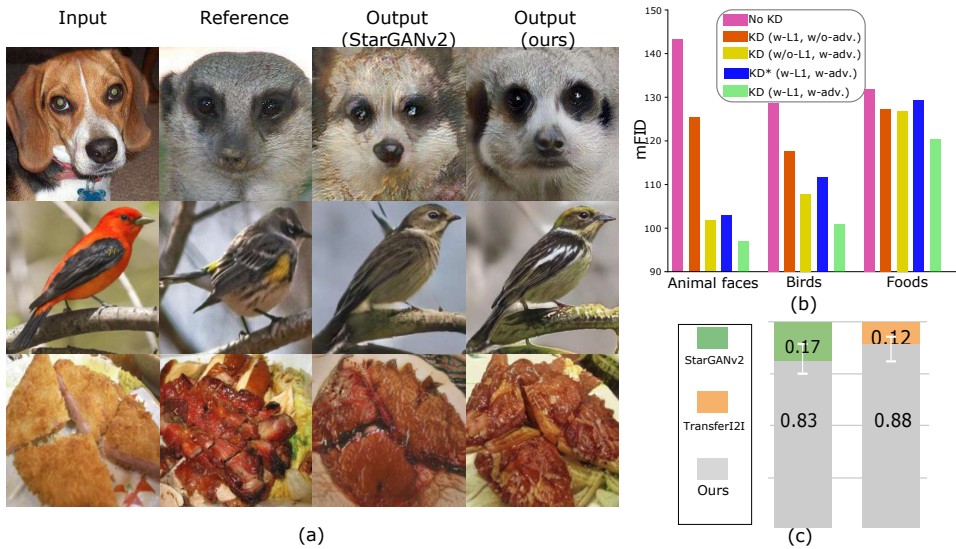

Figure 3: (a) Qualitative comparison on *Animal faces*, *Birds* and *Foods* datasets.(b) Ablation study of variants of our method on *Animal faces*, *Birds* and *Foods*. (c) User study.

| Datasets | Animal faces (10/per class) | | | | Birds (78/per class) | | | | Foods (110/per class) | | | |
|---|---|---|---|---|---|---|---|---|---|---|---|---|
| Method | mKID×100↓ | mFID↓ | RC↑ | FC↑ | mKID×100↓ | mFID↓ | RC↑ | FC↑ | mKID×100↓ | mFID↓ | RC↑ | FC↑ |
| Latent-guided I2I translation | | | | | | | | | | | | |
| SDIT | 31.4 | 283.6 | 5.51 | 4.64 | 22.7 | 223.5 | 8.90 | 8.71 | 23.7 | 236.2 | 11.9 | 11.8 |
| DMIT | 29.6 | 280.1 | 5.98 | 5.11 | 23.5 | 230.4 | 12.9 | 11.4 | 19.5 | 201.4 | 8.30 | 10.4 |
| DRIT++ | 26.6 | 270.1 | 4.81 | 6.15 | 24.1 | 246.2 | 11.8 | 13.2 | 19.1 | 198.5 | 10.7 | 12.7 |
| StarGANv2 | 11.38 | 131.2 | 12.4 | 14.8 | 10.7 | 152.9 | 25.7 | 21.4 | 6.72 | 142.6 | 34.7 | 22.8 |
| DeepI2I | 11.48 | 137.1 | 10.3 | 9.27 | 8.92 | 146.3 | 20.8 | 22.5 | 6.38 | 130.8 | 30.2 | 19.3 |
| TransferI2I | 9.25 | 103.5 | 22.3 | 25.4 | 6.23 | 118.3 | 27.1 | 28.4 | 3.62 | 107.8 | 43.2 | 24.8 |
| Ours | **9.01** | **94.7** | **25.6** | **27.8** | **6.15** | **107.4** | **29.5** | **30.2** | 5.52 | 115.2 | 38.2 | 21.6 |
| StarGANv2* | 10.8 | 119.4 | 30.6 | 35.7 | 6.08 | 125.7 | 29.4 | 38.7 | 5.86 | 115.6 | 43.3 | 24.7 |
| Ours* | **9.97** | **92.8** | **33.4** | **39.9** | **5.88** | **110.4** | **32.5** | **41.3** | **5.28** | **105.9** | **48.5** | **26.1** |
| Reference-guided I2I translation | | | | | | | | | | | | |
| StarGANv2* | 13.2 | 134.7 | 29.3 | 36.6 | 8.53 | 128.7 | 28.6 | 25.5 | 6.38 | 132.2 | 35.6 | 23.6 |
| Ours* | **9.45** | **96.2** | **37.2** | **39.5** | **6.96** | **103.2** | **32.6** | **34.2** | **5.92** | **120.5** | **39.3** | **25.4** |

Table 1: Comparison with baselines. * means the training image resolution is $256 \times 256$.

**Reference-guided I2I translation.** We conduct multi-class I2I translation on three datasets: *Animal faces* (Liu et al., 2019), *Birds* (Van Horn et al., 2015) and *Foods* (Kawano & Yanai, 2014). The *Animal faces* dataset contains 1,490 images and 149 categories in total, *Birds* has 48,527 images and 555 classes , *Foods* consists of 31,395 images and 256 classes. We compare to StarGANv2 (Choi et al., 2020). We also explore a wide variety of configurations for our approach, including: no knowledge distillation (*No KD*), knowledge distillation only using *L1* distance between the teacher generator and the student generator (*KD (w-L1, w/o-adv.)*), knowledge distillation only using the *adversarial* loss (*KD (w/o-L1, w-adv.)*) and knowledge distillation with both losses (*KD (w-L1, w-adv.)*). We also ablate the style-mixed triplets. We replace both the reference image $\mathbf{x_t^2}$ and the style-mixed output $\mathbf{y_t}$ with the input image $\mathbf{x_t^1}$ (indicated by *KD\* (w-L1, w-adv.)*).

*Knowledge distillation and style-mixed triplets.* Figure 3(b) presents a comparison between several variants of our method in terms of mean FID (mFID) on three datasets. Taking Animal faces as an example, adding either *L1* distance or the *adversarial* loss improves the I2I translation task in general compared to the I2I model trained from scratch (mFID: 134.7). Finally combining both losses obtains the best score (mFID: 96.2), indicating that the proposed method largely improves I2I translation performance. Without the style-mixed triplets, the model performance largely drops (mFID:104.1), even with both the $L1$ distance and the adversarial loss. This is probably because the generator fails to perform disentanglement, and the reference encoder does not learn the style information. In conclusion, this experiment validates the effectiveness of the style-mixed triplets.

*Results.* Table 1 (*reference-guided I2I translation*) reports results for baseline and our method on *Animal faces*, *Birds* and *Foods* datasets with image resolution $256 \times 256$. StarGANv2 training from scratch obtains lower results (e.g., mFID:134.7 on *Animal faces* dataset), indicating training I2I model from scratch is challenging when given limited data. Our method achieves a significant advantage (mFID: 96.2 on *Animal faces* dataset), demonstrating that the proposed knowledge transfer

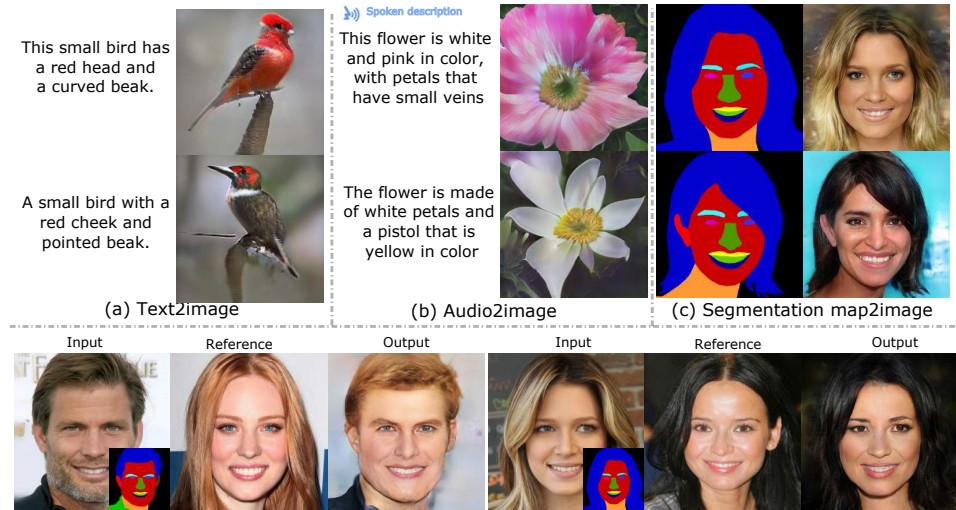

Figure 4: Qualitative results of the proposed method on various image synthesis tasks.

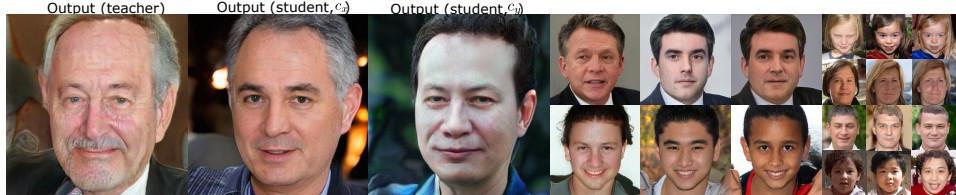

Figure 5: Examples generated after semantic diversity loss. For each triplet, we depict the teacher's output(left), the student's output (middle) with condition $\mathbf{c}_x$ and the one (right) with condition $\mathbf{c}_y$. Note the student model is SSA-GAN. Results could be found in the Appendix C when student model is S2IGAN.

facilitates I2I translation. Figure 3(a) compares the results of mapping the input to the target image domain with a reference image. Given both the input and reference image our method manages to generate higher quality images than StarGANv2, that retain the pose of the input and keep a similar style as the reference image. It clearly indicates that our method improves the image synthesis model. Note that the existing approaches which address transfer learning for I2I (Wang et al., 2020b; 2021) cannot be applied to this case, since they require the I2I image synthesis architecture to be the same as the pretrained GAN architecture.

**Latent-guided I2I translation.** We also validate the proposed method without reference image. We compare to StarGANv2, SDIT (Wang et al., 2019), DRIT++ (Lee et al., 2020b), DMIT (Yu et al., 2019b), DeepI2I and TransferI2I. We use random noise to control the style information of output images. As reported in Table 1 (*latent-guided I2I translation*), all baselines training from scratch (from 4rd to 6th row) suffer a significant disadvantage on small dataset. StarGANv2 obtains the best result (131.2 mFID on *Animal faces*). However, using knowledge transfer TransferI2I performs significantly better (103.5 mFID on *Animal faces*), and achieves the best score on *Foods* dataset. Finally, our method obtains the best results on both *Animal faces* and *Birds* datasets, and competitive performance on the *Foods* dataset. This could be due to the fact that the *Foods* dataset has more training data, and the importance of transfer learning is negligible. We also evaluate our method with higher resolution images (256×256). As shown in the last two rows of Table 1 (*latent-guided I2I translation*), we still retain a large advantage comparing to StarGANv2, demonstrating that our method still obtains better performance for higher resolution images. Neither DeepI2I nor TransferI2I is applicable, since there is no the released pretrained model for usage (they are based on the BigGAN). We conduct a user study and ask subjects to select the results that is *more realistic given the target label, and has the same pose as the input image*. We apply pairwise comparisons (forced choice) with 20 users (100 image pairs/user) for I2I translation. Experiments are performed on images from the Animal faces dataset. Fig. 3(c) shows that our method considerably outperforms the other methods. The synthesized images on three datasets are in the Appendix C.

**text2image**

| Method | FID↓ | KID*100↓ | IS↑ |
|---|---|---|---|
| SSA-GAN | 30.48 | 0.91 | 4.29 |
| Ours(w-L1,w/o-adv.) | 23.74 | 0.80 | 4.19 |
| Ours (w/o-L1,w-adv.) | 23.01 | 0.78 | 4.22 |
| Ours † | 23.26 | 0.79 | 4.26 |
| Ours ‡ | 24.78 | 0.83 | 4.23 |
| Ours | **20.19** | **0.64** | **4.52** |

**audio2image**

| Method | FID↓ | KID*100↓ | IS↑ |
|---|---|---|---|
| S2IGAN | 109.02 | 9.12 | 2.97 |
| Ours(w-L1,w/o-adv.) | 85.74 | 6.48 | 2.94 |
| Ours (w/o-L1,w-adv.) | 80.21 | 5.96 | 3.19 |
| Ours † | 88.81 | 6.52 | 3.03 |
| Ours ‡ | 90.76 | 7.43 | 2.99 |
| Ours | **70.88** | **3.96** | **3.97** |

**SS2image**

| Method | SS2image (w/o reference) | | | | Method | SS2image (w reference) | | | |
|---|---|---|---|---|---|---|---|---|---|
| | FID↓ | KID*100↓ | IS↑ | mIoU↑ | | FID↓ | KID*100↓ | IS↑ | mIoU↑ |
| SPADE | 43.91 | 3.26 | 2.53 | 45.3% | SEAN | 36.57 | 2.34 | 2.67 | 44.7% |
| OASIS | 37.60 | 1.74 | 2.75 | 48.6% | - | - | - | - | - |
| CoCosNetv2 | 36.31 | 1.69 | 2.71 | 46.3% | - | - | - | - | - |
| Ours † | **32.98** | **1.42** | **2.80** | **50.4%** | Ours † | **27.17** | **1.38** | **2.89** | **52.1%** |
| Ours ‡ | 37.49 | 1.86 | 2.66 | 47.3% | Ours ‡ | 34.26 | 2.30 | 2.68 | 46.2% |
| Ours | 35.32 | 1.76 | 2.72 | 48.4% | Ours | 35.70 | 2.31 | 2.69 | 45.9% |

Table 2: The results on a wide range of image synthesis tasks. † means the condition **c** is **1** in the student generator and **0** in the student discriminator. ‡ means we remove the semantic diversity loss (Eq. 7) from final objective (Eq. 8). Left: text-to-image. Middle: audio-to-image. Right: semitic segmentation (SS) to image.

## 4.2 DATA-FREE KNOWLEDGE DISTILLATION FOR X2I TRANSLATION

We apply our approach for conditional image synthesis. We consider conditioning information **c** from text, audio and segmentation maps. We explore four variants of our method: knowledge distillation only using *L1* distance between the teacher generator and the student generator (*Ours (w-L1, w/o-adv.)*); knowledge distillation only using the *adversarial* loss (*Ours (w/o-L1, w-adv.)*); next we set the condition **c=1** in the student generator and **0** in the student discriminator (*ours* †); finally we remove the semantic diversity loss in Eq. 7 from the final objective in Eq. 8 (*ours* ‡). More details on training and hyperparameters can be found in Appendix A.

**Text-to-image.** We evaluate the proposed method on the *CUB* bird dataset (Welinder et al., 2010). It contains 8,855 training images (150 species) and 2,933 test images (50 species), following 10 text descriptions for each bird. Here we use 10 images per class for training, and verify our method on the test dataset. We compare with SSA-GAN (Hu et al., 2021) which is one of state-of-the-art methods. Results are depicted in Figure 1, Figure 4 (a) and Table 2 (left).

**Audio-to-image.** We evaluate on the commonly-used dataset: *Oxford-102* (Nilsback & Zisserman, 2008), where 82 categories and 10 images per category are selected for training, and 20 categories and 1,155 images for test. Note that we split the training and test categories following S2IGAN (Wang et al., 2020a). Results are depicted in Figure 1, Figure 4 (b) and Table 2 (middle).

**Semantic segmentation-to-image.** Here we conduct experiments by conditioning on segmentation map on the *CeleAMask-HQ* (Lee et al., 2020a) dataset containing 30,000 image and segmentation mask pairs. We follow the same train/test split as SEAN (Zhu et al., 2020b). We randomly select 500 pairs of data from the train set for training, and 2,000 pairs for testing. To evaluate our method, we compare to three baselines: SPADE (Park et al., 2019), OASIS (Simonyan et al., 2013) and CoCosNetv2 (Zhou et al., 2021), which are trained from scratch. We report the generator performance using the pretrained StyleGAN optimized on AFHQ (Choi et al., 2020)(animal face dataset). Results are depicted in Figure 1, Figure 4 (c) and Table 2 (right, w/o reference). We also compare to SEAN (Zhu et al., 2020b) which additionally uses a reference image to control the style of the output image, the results are shown in Figure 4 (d) and Table 2 (right, w reference).

As reported in Table 2, our approach outperforms all baselines without using knowledge distillation on several metrics. The semantic diversity loss gets the best performance except for the semantic-map-to-image task. The qualitative results in Figure 1 and 4 indicate that our method can generate high-quality images with limited data due to the benefit of the knowledge transferred from the expert teacher network. Figure 5 shows the output images of the X2I generator after knowledge distillation and shows that we synthesize diverse outputs given different condition **c**. In conclusion, our method is an efficient, general purpose mechanism for conditional image synthesis with limited data.

## 5 CONCLUSIONS

Conditional image synthesis tasks require vast amounts of training data. Therefore, we proposed to transfer knowledge from a unconditional pretrained GAN model to conditional image synthesis tasks. We perform the distillation with style-mixed triplets that can be automatically computed from high-quality GANs. Our experiments confirmed that the proposed transfer learning method obtains state-of-the-art results and allows to train high-quality X2I systems from fewer labeled samples.

**Acknowledgement.** We acknowledge the support from Huawei Kirin Solution, and the Spanish Ministerio de Ciencia, Innovacion, y Universidades for funding the project PID2019-104174GB-I00, 10.13039/501100011033.

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

## A    ARCHITECTURE AND TRAINING DETAILS

The proposed method is implemented in Pytorch (Paszke et al., 2017). We transfer knowledge from StyleGAN to X2I translation. We introduce here the architecture and model details for each task. Note we perform the knowledge distillation on one GPU(Quadro RTX6000) with 24GB VRAM.

### A.1    KNOWLEDGE DISTILLATION FOR TEXT-TO-IMAGE TRANSLATION

For text-to-image translation the training is composed of two stages: data-free distillation from the unconditional prerained StyleGAN to text-to-image translation model, and finetuning text-to-image translation model using the target data and label.

**Model details for the first stage.** The student model is the same as SSA-GAN (Hu et al., 2021), which is composed of three sub-networks: a text encoder, a student generator $G_S$, a student discriminator $D_S$. The text encoder is the pretrained one provided by (Xu et al., 2017). The generator takes random noise as input, following one fully connection ($100 \times 8192$) and 7 ResBlocks [5]. The input feature size of the first ResBlock is $B \times 512 \times 4 \times 4$ ($B$ is the batch size). We extract the hierarchical representation from StyleGAN (Karras et al., 2019). The first feature from StyleGAN is $B \times 512 \times 8 \times 8$, and fed into the student generator, which is summed with the student generator feature representation which has the same dimension. The student discriminator contains one convolution layer and 6 ResBlocks.

We optimize the model using Adam (Kingma & Ba, 2014) with batch size of 16. The learning rates of the generator and the discriminator are set as 0.0001 and 0.0004 with exponential decay rates of $(\beta_1, \beta_2) = (0.0, 0.9)$. The model is trained for 300 epochs for knowledge distillation. In the second stage, we iterate 300 epochs, with same batch size, learning rate and exponential decay rate. In Eq. 6 both $\alpha_l$ and $\beta$ are identical. For the specific features of which the dimension is less than 128, we set them 0.1. In other case they are 0.01. In Eq. 8 $\lambda_{adv}$ and $\lambda_{kdl}$ are 1, and $\lambda_{srl}$ is 0.1.

**Model details for the second stage.** We finetune the well-initialized student network learned in the first stage. In this stage, we get access to the target data. We keep all the details on training and the hyperparameters of SSA-GAN (Hu et al., 2021). We refer readers for more detailed information to (Hu et al., 2021).

---

[5]After each ResBlock the feature resolution is half of the previous one in both encoder and discriminator, and two times in generator.

## A.2 Knowledge Distillation for Audio-to-Image Translation

For audio-to-image translation the training is composed of two stages: data-free distillation from the unconditional prerained StyleGAN to audio-to-image translation model, and finetuning audio-to-image translation model using the target data and label.

**Model details for the first stage.** We perform knowledge distillation for audio-to-image based on S2IGAN (Wang et al., 2020a). To obtain stable training, S2IGAN proposed a progressive mechanism to generate multi-scale images. S2IGAN introduced a two-stage study, i.e., a speech semantic embedding stage and an image generation stage. The former is to extract the speech embedding with a pretrained speech extractor, the latter takes as an input both random noise and the speech embedding to synthesize photo-realistic images in a multi-stage (coarse-to-fine) way. The generator is composed of one fully connected layer ($228 \times 2048$), one batchnorm, one Gaussian Error Linear Unit(GLU) (Devlin et al., 2019) and 7 UpsBlocks [6]. The first UpsBlock is $B \times 512 \times 4 \times 4$ ($B$ is the batch size). We extract the hierarchical representation from StyleGAN (Karras et al., 2019). The first feature from StyleGAN is $B \times 512 \times 8 \times 8$, and fed into the student generator, which is summed with the student generator feature representation which has the same dimension.

We optimize the model using Adam (Kingma & Ba, 2014) with batch size of 12. The learning rates of both the generator and the discriminator are set as 0.0002 with exponential decay rates of $(\beta_1, \beta_2) = (0.5, 0.999)$. The model is trained for 300 epochs for knowledge distillation. In the second stage, we iterate 300 epochs, with same batch size, learning rate and exponential decay rate. In Eq. 6 both $\alpha_l$ and $\beta$ are identical, and are 10. In Eq. 8 $\lambda_{adv}$, $\lambda_{kdl}$ and $\lambda_{srl}$ are 10.

**Model details for the second stage.** We finetune the well-initialized student network learned in the first stage. In this stage, we get access to the target data. We keep all the details on training and the hyperparameters of S2IGAN (Wang et al., 2020a). We refer readers to check more detailed information to (Wang et al., 2020a).

## A.3 Knowledge Distillation for Segmentation Map-to-Image Translation

For segmentation map-to-image translation the training is composed of two stages: data-free distillation from the unconditional prerained StyleGAN to segmentation map-to-image translation model, and finetuning segmentation map-to-image translation model using the target data and label.

**Model details for the first stage.** For segmentation map-to-image task, we use SEAN (Zhu et al., 2020b) as the student model, since it is state-of-the-art. SEAN is composed of one generator, one style encoder and one discriminator. The generator contains several SEAN ResBlks. Each of the ResBlks is followed by a nearest neighbor upsampling layer. We extract the hierarchical representation from StyleGAN (Karras et al., 2019). The first feature from StyleGAN is $B \times 512 \times 8 \times 8$.

We use the pretrained StyleGAN optimized on HHFQ human face (Karras et al., 2019) except for semantic segmentation-to-image(SS2image), which is on AFHQ dataset (Choi et al., 2020). In fact we are able to use the pretrained one from HHFQ, which further improve the performance.

In the knowledge distillation stage, we set the learning rates to 0.0001 and 0.0004 for the generator and discriminator, respectively (Heusel et al., 2017). For the optimizer, we choose Adam (Kingma & Ba, 2014) with $\beta_1 = 0, \beta_2 = 0.999$, with batch size 4. In second stage (segmentation map-to-image translation), we use same setting except for batch size (i.e., 8). We update 100 epochs for knowledge distillation. In second stage (image-to-image translation) we iterate 100 epochs. In Eq. 6 both $\alpha_l$ and $\beta$ are identical. For the specific features of which the dimension is less than 128, we set them 0.1. In other case they are 0.01. In Eq. 8 $\lambda_{adv}$ and $\lambda_{kdl}$ are 1, and $\lambda_{srl}$ is 0.1.

**Model details for the second stage.** We finetune the well-initialized student network learned in the first stage. In this stage, we get access to the target data. We keep all the details on training and

---

[6]Each UpsBlock consists of one interpolate with scale factor 2, one convolutional layer ($3 \times 3$ kernal size, stride 1), batchnorm and GLU (Devlin et al., 2019). After each UpsBlock, the resolution is half of the previous one in the discriminator, and two times the previous one in generator.

the hyperparameters of SEAN (Zhu et al., 2020b). We refer readers to check all detailed information in (Zhu et al., 2020b).

### A.4 KNOWLEDGE DISTILLATION FOR IMAGE-TO-IMAGE TRANSLATION

**Model details.** We take StarGANv2 (Choi et al., 2020) as the student model. StarGANv2 contains 5 sub-networks: a context encoder, a mapping network, a style encoder, a generator and a discriminator. For both content encoder and generator, we extract hierarchical representations with dimension size range from 16 to 256. We also extract the corresponding dimension hierarchical representations from the pretrained generator and discriminator respectively. To align the student style encoder with the teacher discriminator, we extract hierarchical representations for both networks with the dimension size ranging from 4 to 256.

We use the Adam (Kingma & Ba, 2014) with $\beta_1 = 0, \beta_2 = 0.99$, with batch size 2. We update 20,000 iterations for knowledge distillation. In second stage (image-to-image translation), we iterate 100,000.

**Model details for the second stage.** We finetune the well-initialized student network learned in the first stage. In this stage, we get access to the target data. We keep all the details on training and the hyperparameters of StarGANv2 (Choi et al., 2020). We refer readers to check all detail information in (Choi et al., 2020).

We are able to explicitly control domains/classes as starGANv2 does. For the latent-guided synthesis, starGANv2 introduces a class-specific mapping networks (one for each class) to project the class embedding into the shared latent space. Thus starGANv2 uses the class-specific mapping network to control the domains/classes. In this paper, we follow the same setup as starGANv2. In the first stage of our method, we only learn one mapping network, which is duplicated to initialize all the mapping networks (one for each class) of the second stage. Then we use the target dataset (second stage) to train the well-initialized class-specific mapping networks.

## B I2I TRANSLATION

### B.1 TWO-CLASS I2I TRANSLATION RESULTS

To evaluate the generalization of our method, here we validate the proposed algorithm for two-class I2I translation on a two-category dataset: cat2dog-200 (Wang et al., 2021) with an image size of $256 \times 256$. In cat2dog-200, the training set is composed of 200 images (100 images/per class) and the test set has 200 images (100 images/per class). We investigate two cases of I2I translation: the style representation is from noise (i.e., latent-guided synthesis) and from the reference image (i.e., reference-guided synthesis). As reported Tables 3 and 4, compared to the transfer learning methods (i.e., DeepI2I (Wang et al., 2020b) and TransferI2I (Wang et al., 2021)) we still maintain a large advantage.

### B.2 MULTI-CLASS I2I TRANSLATION RESULTS

We also further validate the proposed algorithm for multi-class I2I translation on AFHQ-500 (Choi et al., 2020) with an image size of $256 \times 256$. In AFHQ-500, the training set is composed of 500 images (100 images/per class) and the test set has 1500 images (500 images/per class). We investigate two cases of I2I translation: the style representation is from noise (i.e., latent-guided synthesis) and from the reference image (i.e., reference-guided synthesis). As reported Tables 5, compared to StarGANv2 we still maintain a large advantage.

### B.3 VARIANTS OF THE BASELINES

We adapt both DeepI2I (Wang et al., 2020b) and TransferI2I (Wang et al., 2021) to use a very similar architecture to StarGANv2 (the architectural details are in Table 5, 6, 7 of StarGANv2 (Choi et al., 2020)). Specially, to devise the generator, we preserve the subnets of the discriminator of the pretrained StyleGAN as image encoder which has $16 \times 16 \times 512$output, then additionally add 4 ResBlks like StarGANv2, and finally use the subnets of the generator of the pretrained StyleGAN as

| | dog2cat | | cat2dog | |
|---|---|---|---|---|
| | FID | KID*100 | FID | KID*100 |
| DeepI2I | 154.6 | 6.97 | 194.6 | 24.53 |
| TransferI2I | 137.2 | 6.48 | 182.1 | 24.14 |
| Ours | 60.17 | 4.83 | 86.4 | 5.46 |

Table 3: The metric results of *reference-guided synthesis* on *cat2dog-200* dataset. Note we multiply 100 for *KID*.

| | dog2cat | | cat2dog | |
|---|---|---|---|---|
| | FID | KID*100 | FID | KID*100 |
| DeepI2I | 83.71 | 4.26 | 112.4 | 5.67 |
| TransferI2I | 55.2 | 3.97 | 83.6 | 4.56 |
| Ours | 42.7 | 3.46 | 74.2 | 4.03 |

Table 4: The metric results of *Latent-guided synthesis* on *cat2dog-200* dataset. Note we multiply 100 for *KID*.

the decoder to output the image. We keep the mapping network of the pretrained StyleGAN for the I2I translation generator when performing latent-guided I2I translation. We adapt the discriminator of the pretrained StyleGAN as the style encoder when performing reference-guided I2I translation. Similar to StarGANv2, we also use a multitask discriminator, which consists of multiple output branches. We use cat2dog-200 dataset to compare the adapted DeepI2I (DeepI2I*), adapted TransferI2I (TransferI2I* ) and ours. As reported in Tables 6 and 7, compared to the transfer learning methods (i.e., Adapting DeepI2I and TransferI2I ) we still maintain a large advantage.

### B.4 ABLATION OF THE ENCODER

In this paper we use the pre-defined discriminator to initialize the encoder, since the discriminator of the StyleGAN is trained on HHFQ with 70k images, which optimizes it to be an effective feature extractor. A similar technique was also explored in SGD (Shocher et al., 2020) and transferI2I.

To verify that the previously reported results also hold for our method, we also train both the encoder and the reference encoder from scratch in reference-guided translation on Animal faces dataset (10/per class). we achieve 130 of mFID, which is lower than our method (mFID 96.2) . If we train the encoder from scratch, the training suffers from overfitting with limited training images.

## C ADDITIONAL RESULTS

### C.1 DIVERSE OUTPUTS

Figure 6 shows some examples of the synthesized images on three datasets. Taking *Animal faces* as example, given the target class label our method manages to generate high visual quality images.

### C.2 KNOWLEDGE DISTILLATION FOR I2I SYSTEM IN THE FIRST STAGE

Figures 7 and 8 show examples generated after transfer learning on I2I system. This shows that our method successfully distills the style-mixing characteristics from the teacher generator to the student generator.

### C.3 KNOWLEDGE DISTILLATION FOR X2I SYSTEM IN THE FIRST STAGE

Figures 9 and 10 show examples generated after semantic diversity loss. For each triplet, we depict the teacher's output(left), the student's output (middle) with condition $\mathbf{c}_x$ and the one (right) with

| | AFHQ-500(Latent-guided synthesis) | | AFHQ-500(reference-guided synthesis) | |
|---|---|---|---|---|
| | FID | LPIPS | FID | LPIPS |
| StarGANv2 | 40.21 | 0.45 | 41.62 | 0.42 |
| Ours | **35.36** | **0.48** | **34.75** | **0.45** |

Table 5: The metric results on *AFHQ-500* dataset.

| | dog2cat | | cat2dog | |
|---|---|---|---|---|
| | FID | KID*100 | FID | KID*100 |
| DeepI2I* | 189.3 | 15.4 | 205.6 | 29.2 |
| TransferI2I* | 182.6 | 16.7 | 198.4 | 26.7 |
| Ours | 60.17 | 4.83 | 86.4 | 5.46 |

Table 6: The metric results of *Reference-guided synthesis* on *cat2dog-200* dataset. Note we multiply 100 for *KID*.

condition $\mathbf{c}_y$. This shows that our semantic diversity loss manages to diversify the output given the input condition $\mathbf{c}$.

## C.4 ADDITIONAL RESULTS

We additionally depict the generated images in Figures 12- 24.

|  | dog2cat | | cat2dog | |
|---|---|---|---|---|
|  | FID | KID*100 | FID | KID*100 |
| DeepI2I* | 210.7 | 20.7 | 196.5 | 21.8 |
| TransferI2I* | 197.8 | 19.4 | 182.4 | 20.7 |
| Ours | 42.7 | 2.46 | 74.2 | 4.03 |

Table 7: The metric results of *Latent-guided synthesis* on *cat2dog-200* dataset. Note we multiply 100 for *KID*.

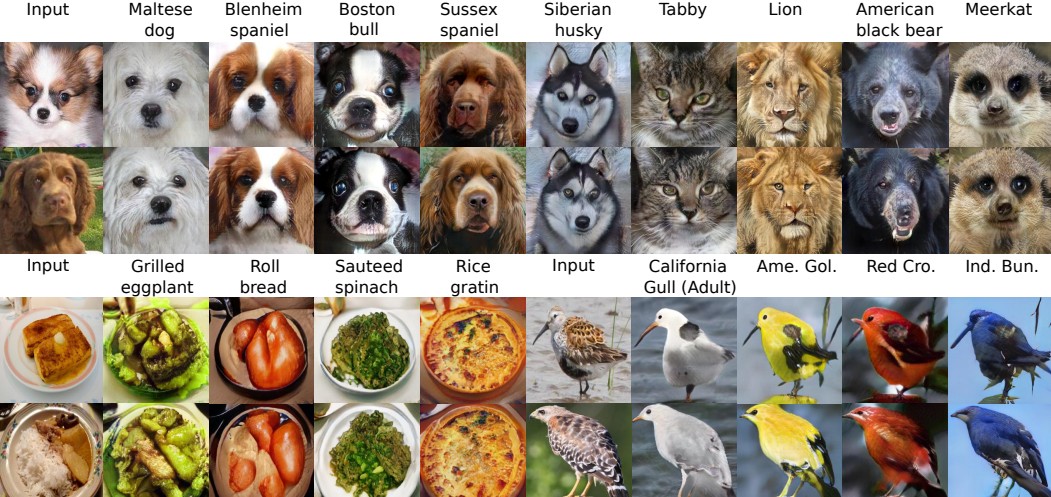

Figure 6: Qualitative results of the proposed method on multi-class I2I translation task. Ame.Gol.:American Goldfinch (Breeding Male), Red Cro.:Red Crossbill (Adult Male), Ind. Bun.: Indigo Bunting (Adult Male).

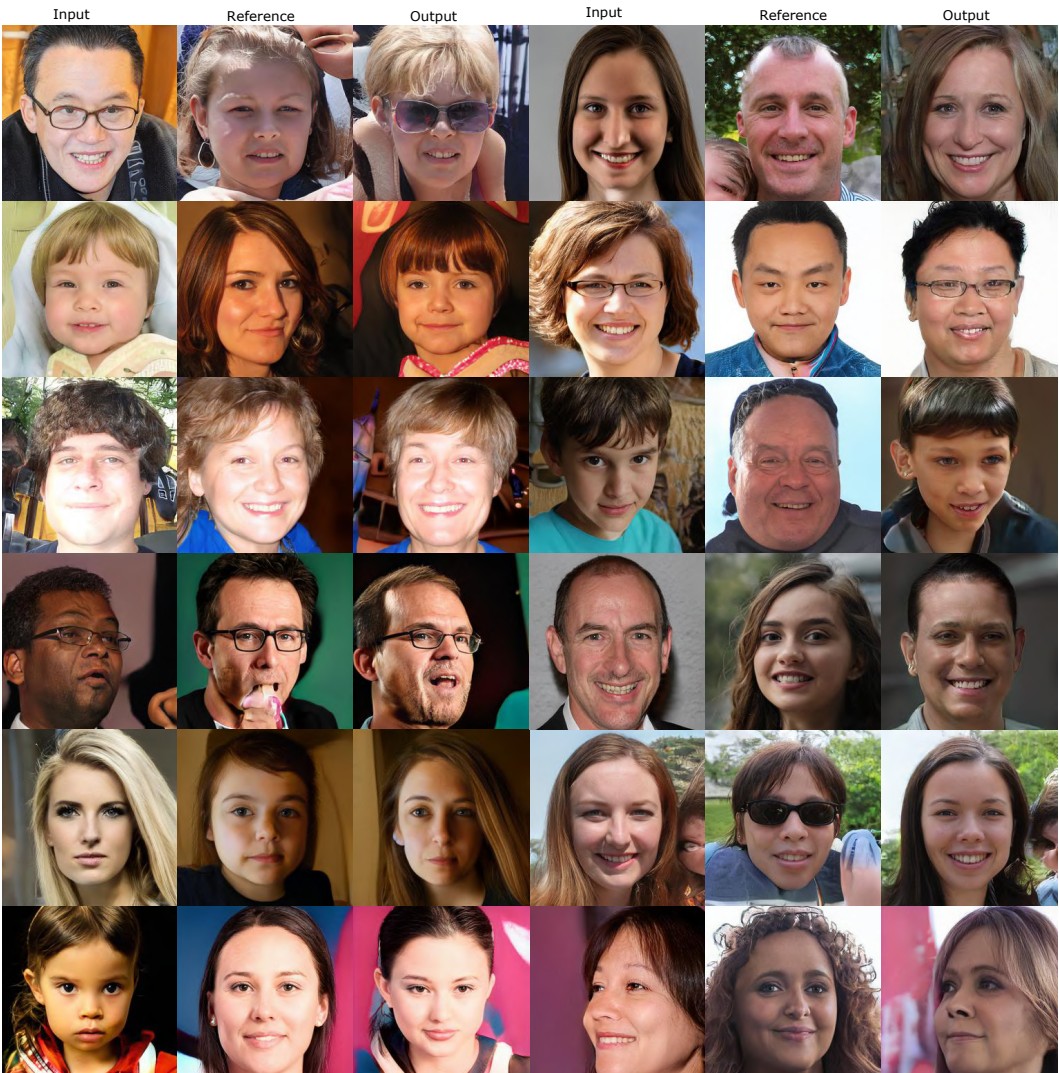

Figure 7: Examples generated after transfer leaning for I2I system. The student model is StarGANv2. Note it is under reference-guided mode.

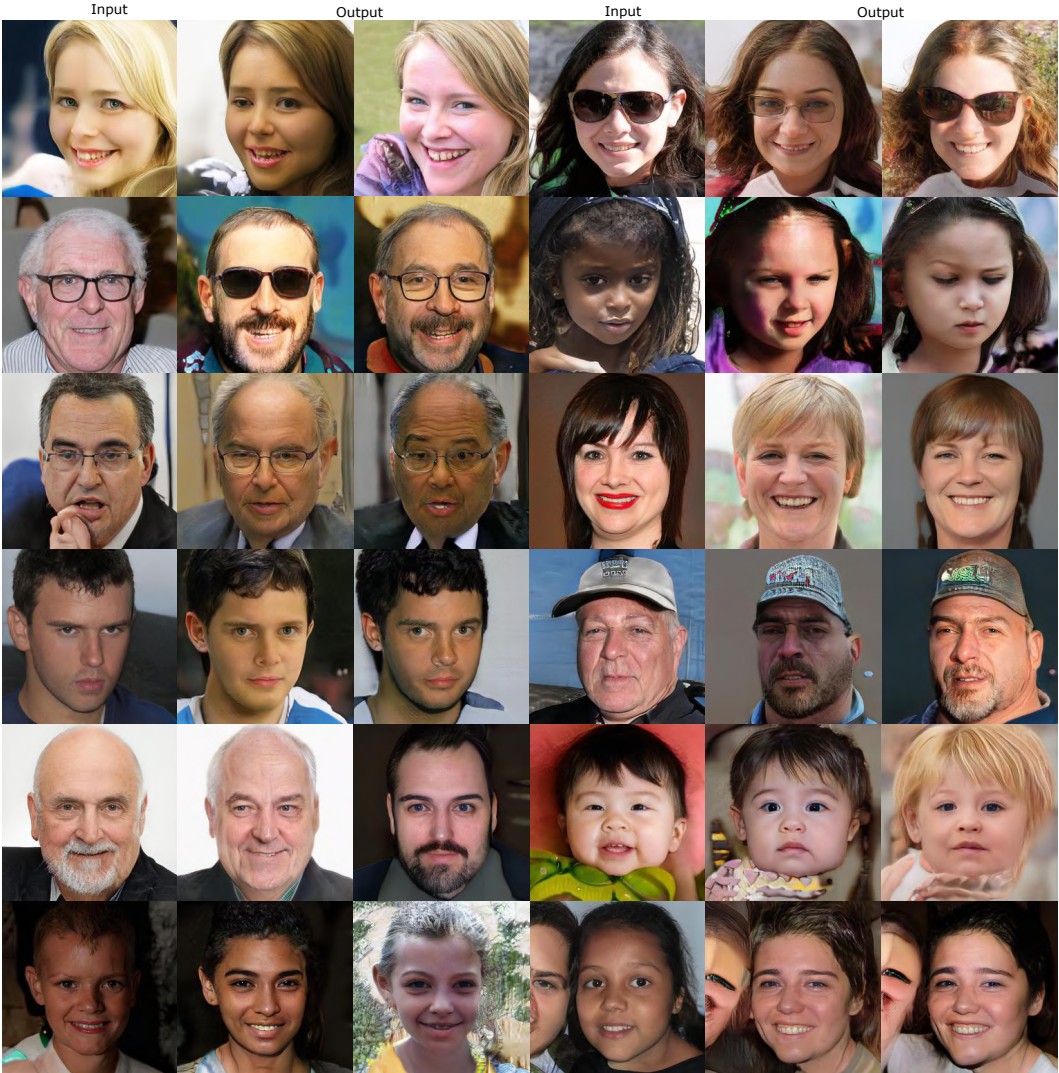

Figure 8: Examples generated after transfer leaning for I2I system. For one input, we sample two noises to generate images. The student model is StarGANv2. Note it is under latent-guided mode.

Output (teacher)    Output (student,$c_x$)    Output (student,$c_y$)    Output (teacher)    Output (student,$c_x$)    Output (student,$c_y$)

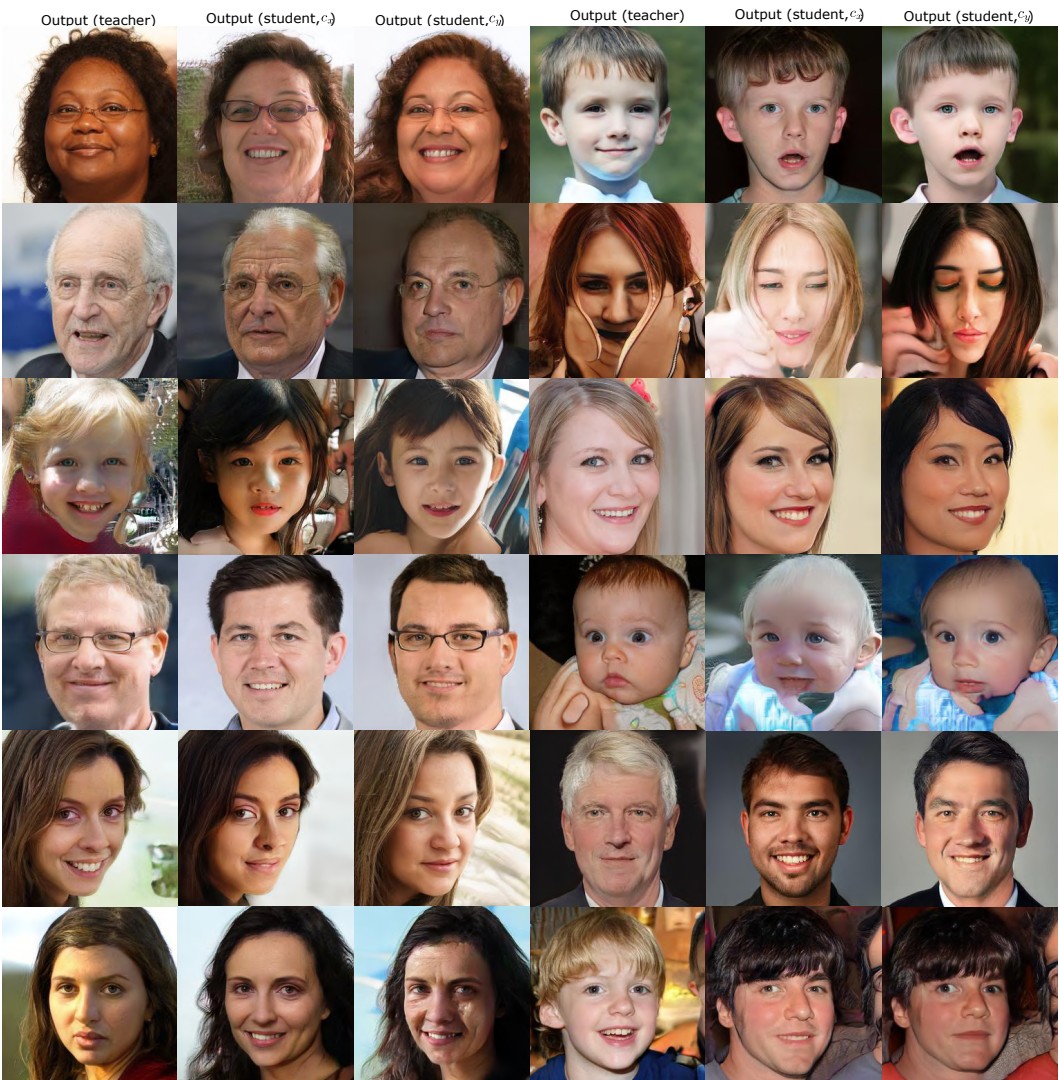

Figure 9: Examples generated after semantic diversity loss. For each triplet, we depict the teacher's output(left), the student's output (middle) with condition $\mathbf{c}_x$ and the one (right) with condition $\mathbf{c}_y$. Note the student model is SSA-GAN.

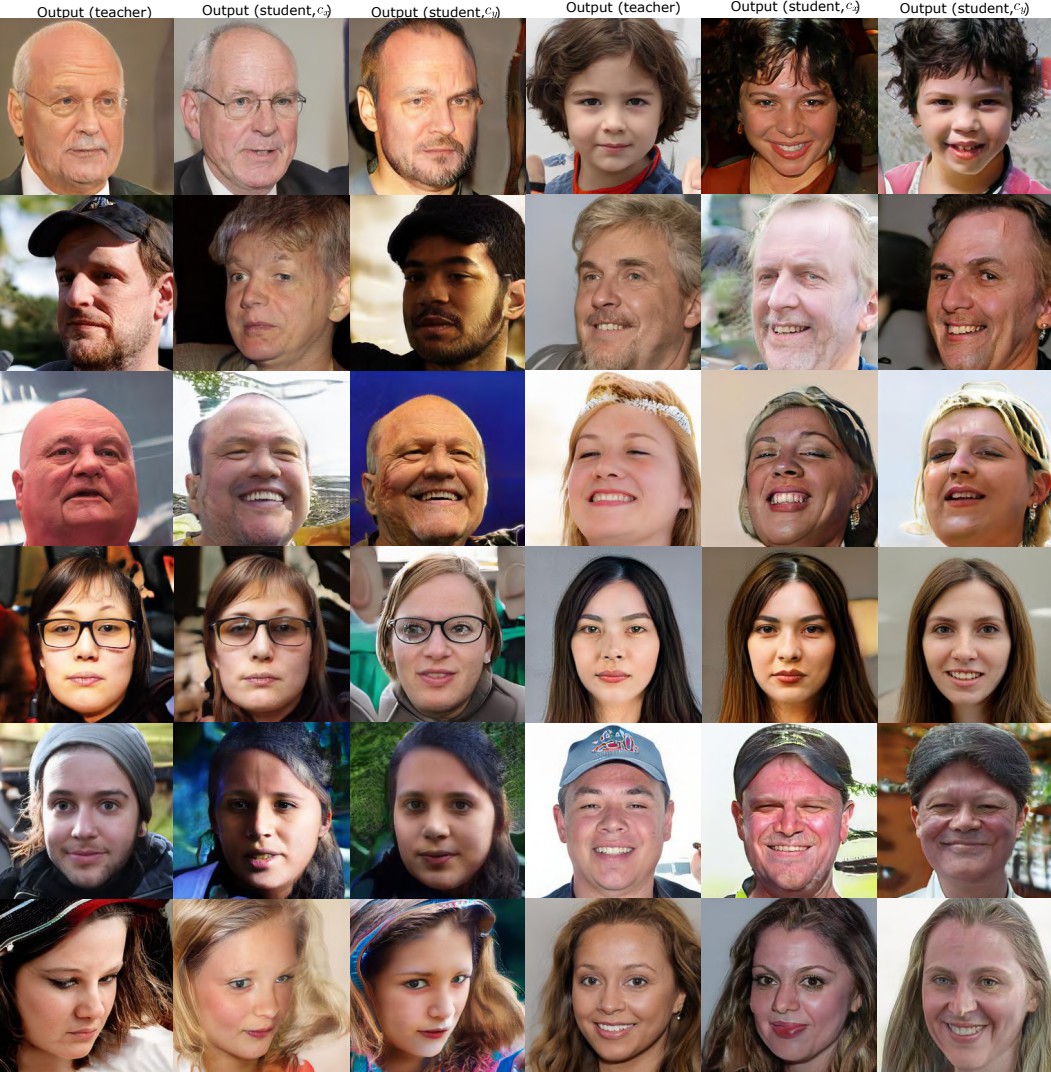

Figure 10: Examples generated after semantic diversity loss. For each triplet, we depict the teacher's output(left), the student's output (middle) with condition $\mathbf{c}_x$ and the one (right) with condition $\mathbf{c}_y$. Note the student model is S2IGAN.

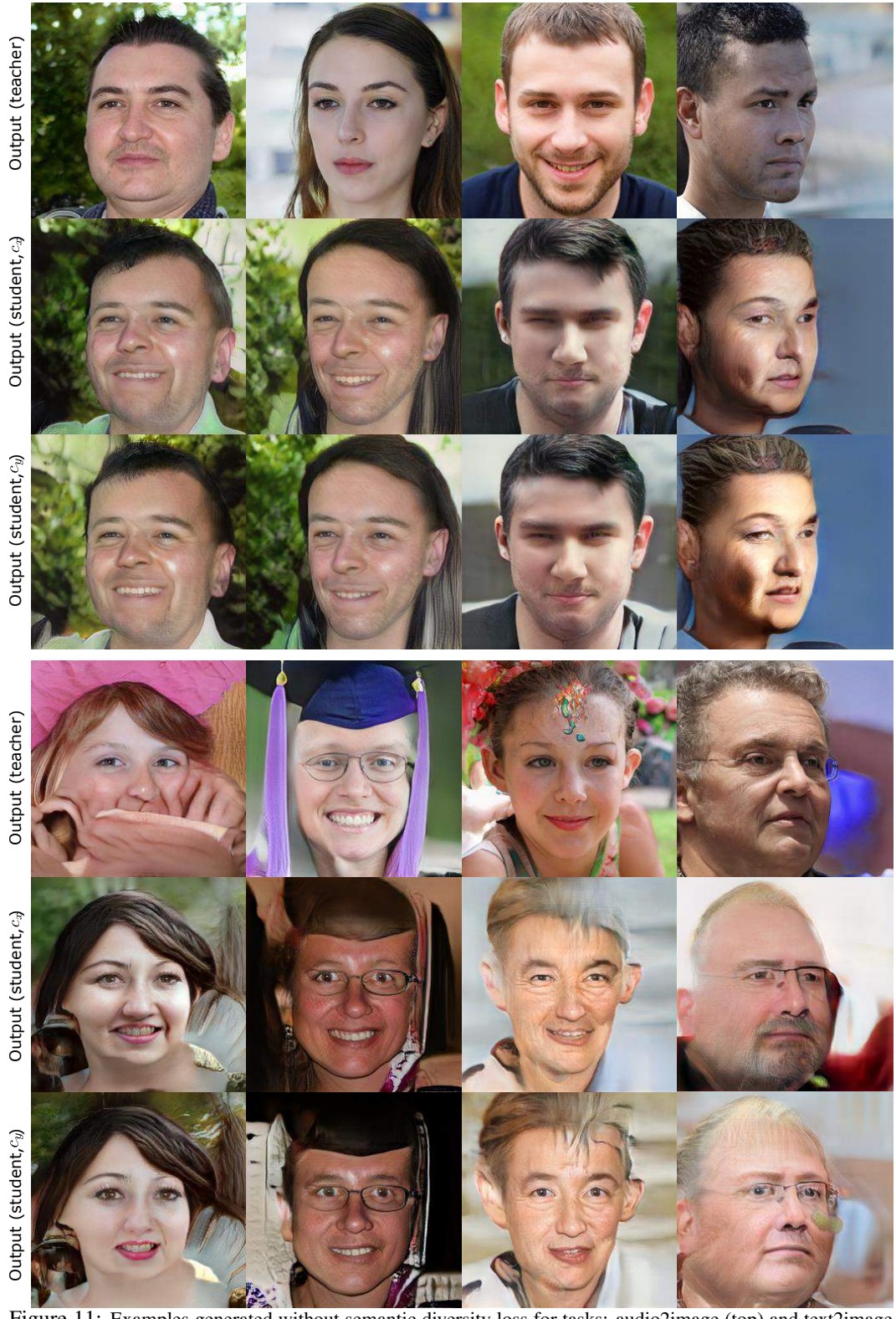

Figure 11: Examples generated without semantic diversity loss for tasks: audio2image (top) and text2image (bottom). For each triplet, we depict the teacher's output, the student's output with condition $\mathbf{c}_x$ and the one with condition $\mathbf{c}_y$.

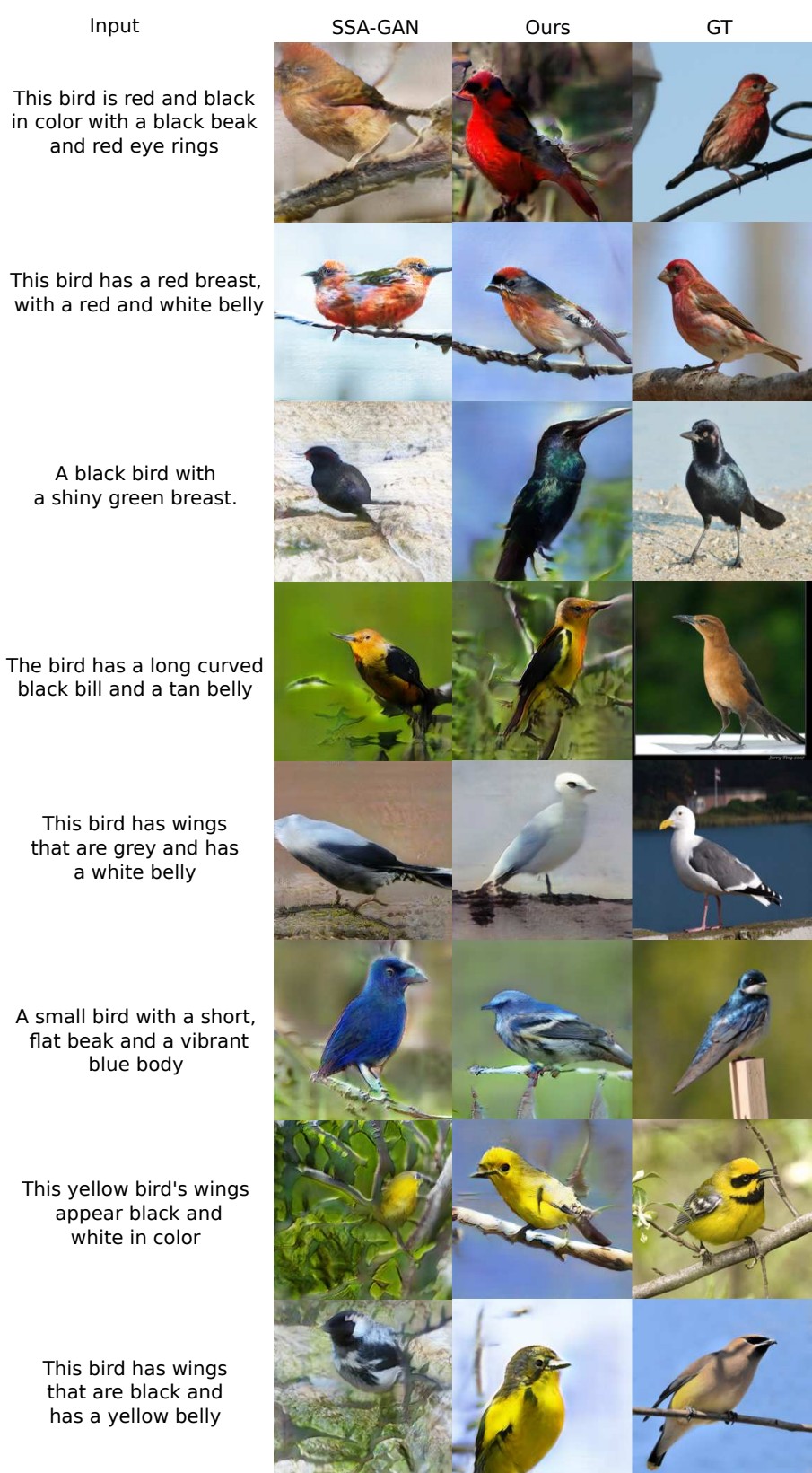

Figure 12: Qualitative results on text-to-image synthesis task.

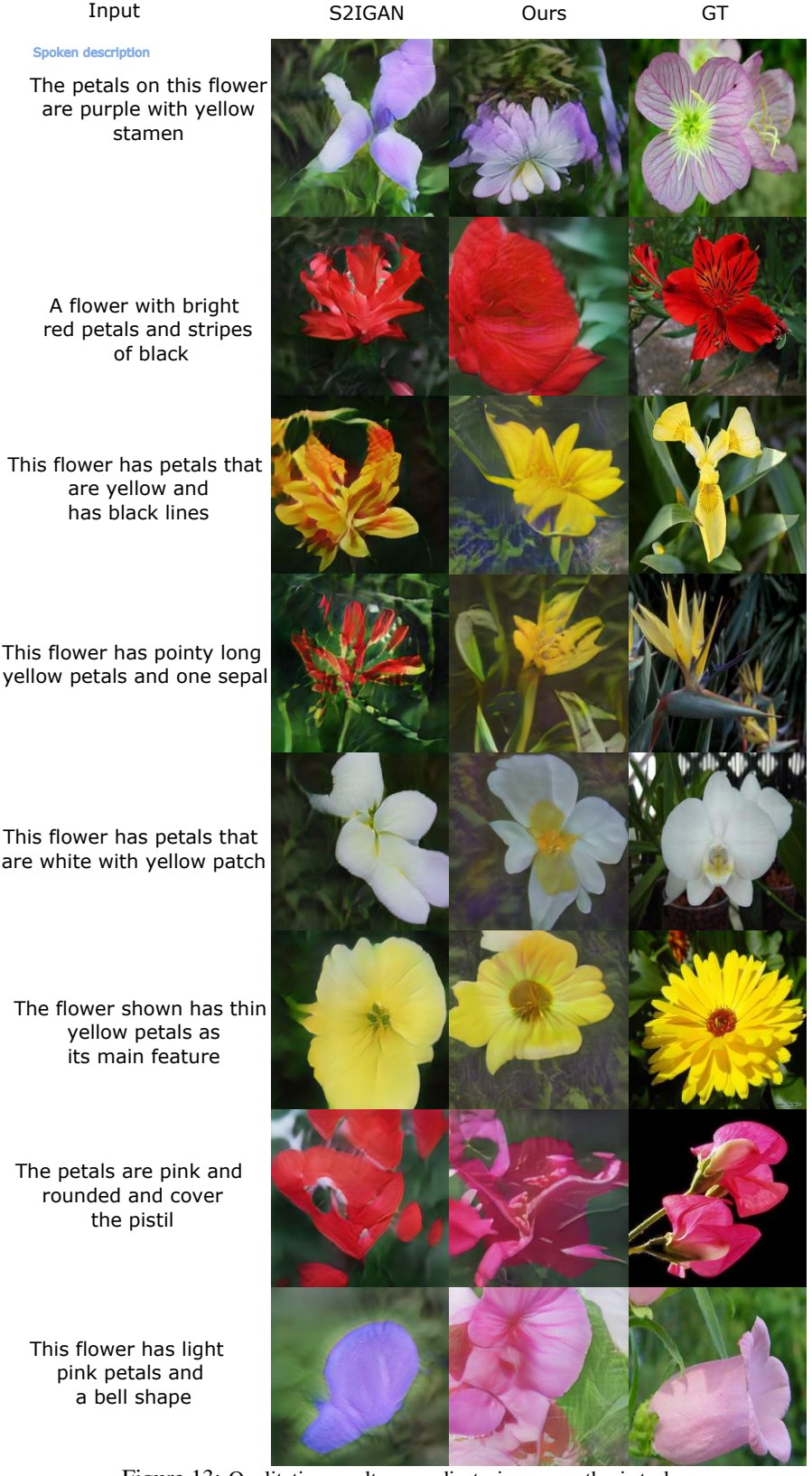

Figure 13: Qualitative results on audio-to-image synthesis task.

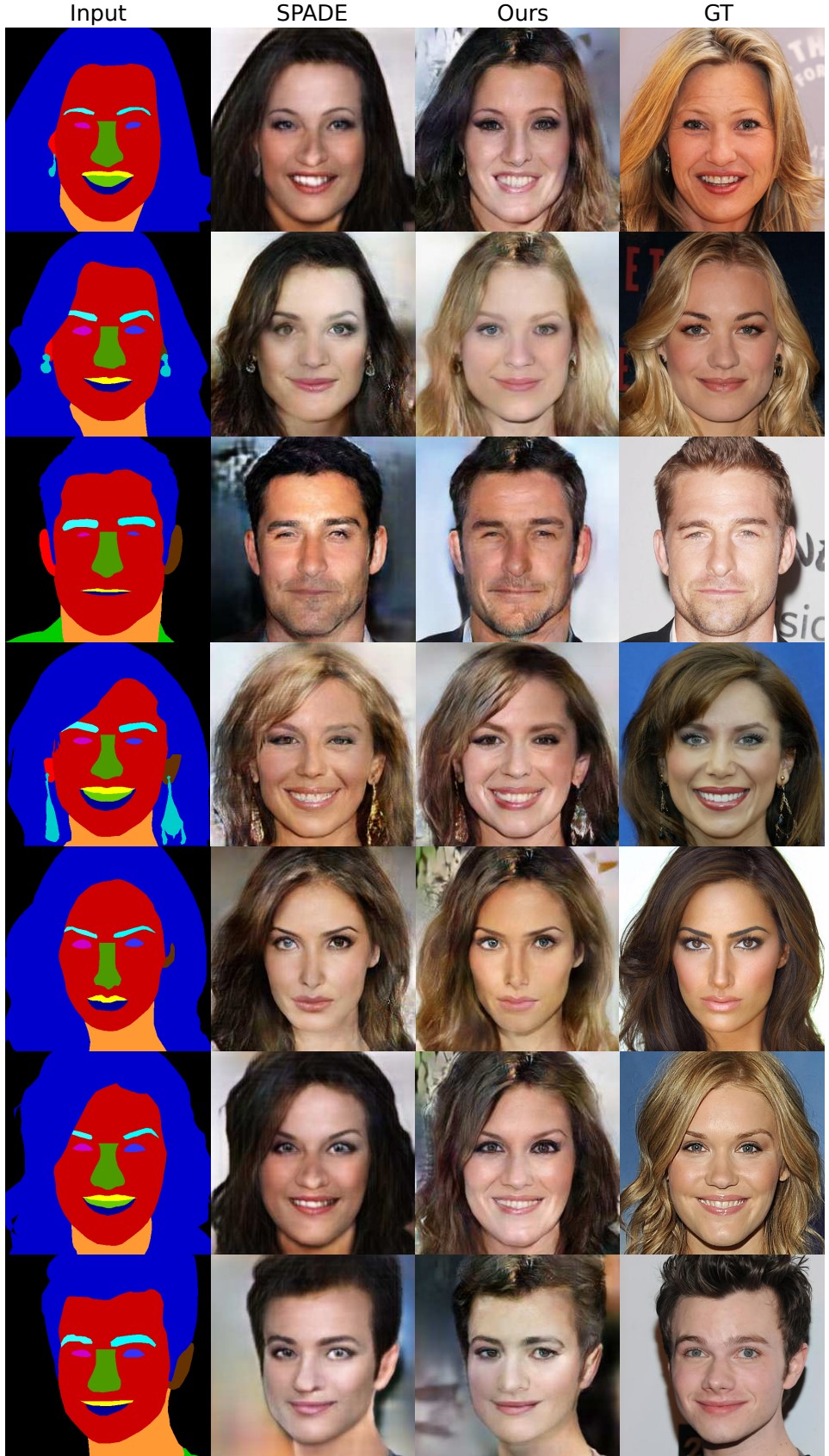

Figure 14: Qualitative results on segmentation map-to-image synthesis task.

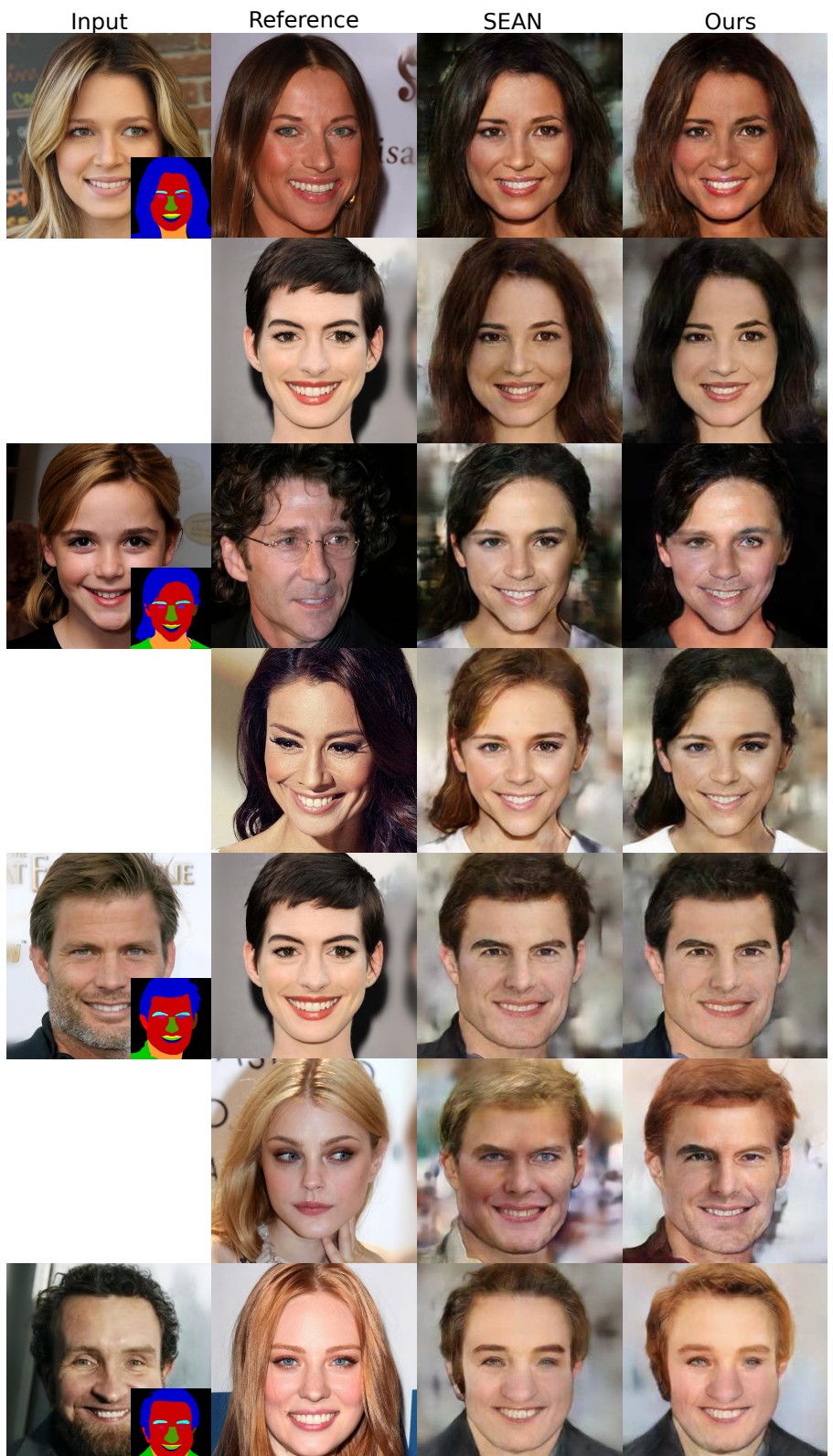

Figure 15: Qualitative results with reference images on segmentation map-to-image synthesis task.

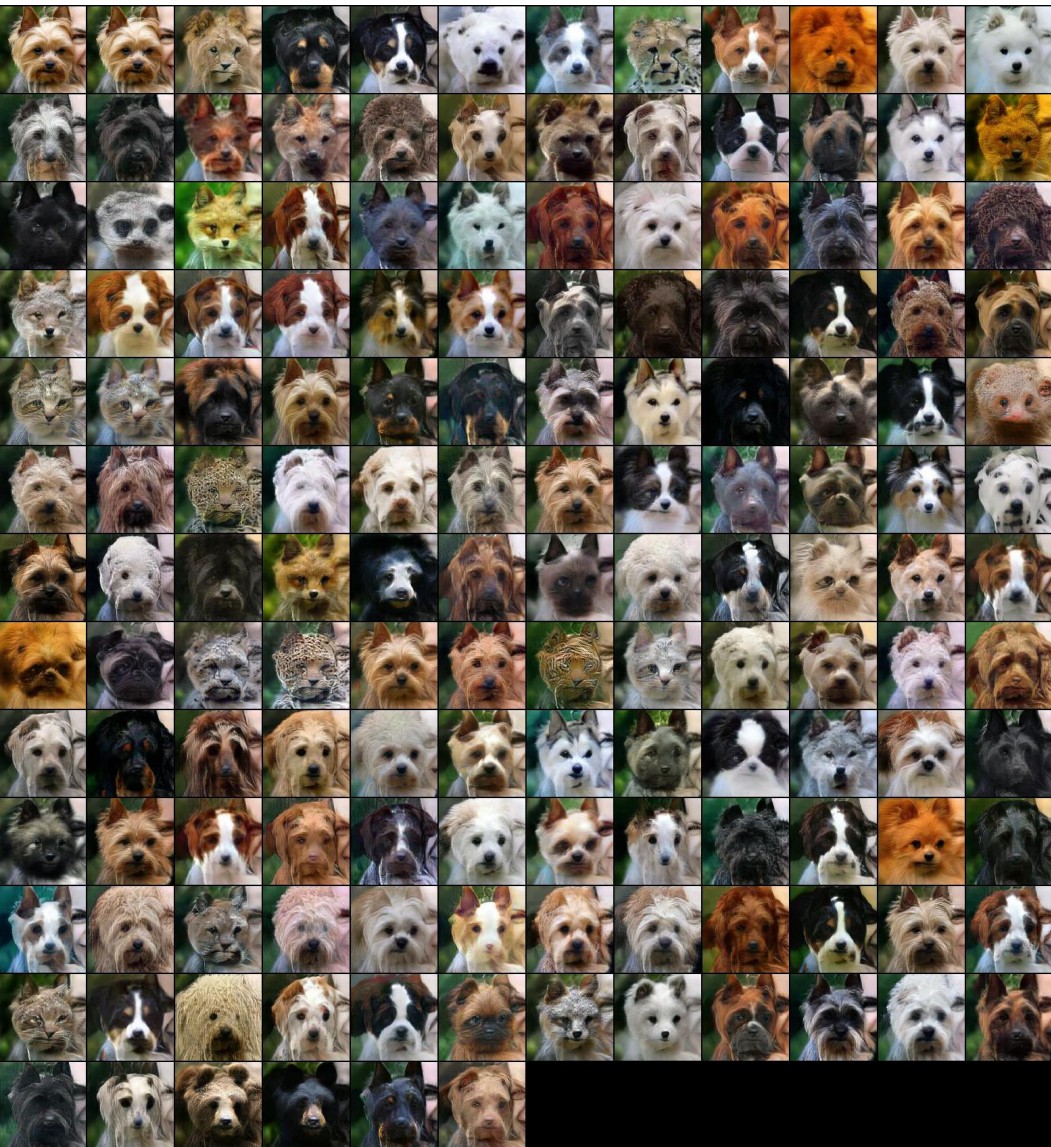

Figure 16: Qualitative results on image-to-image synthesis task. We translate the input image (top left) into all 149 categories. Please zoom-in for details.

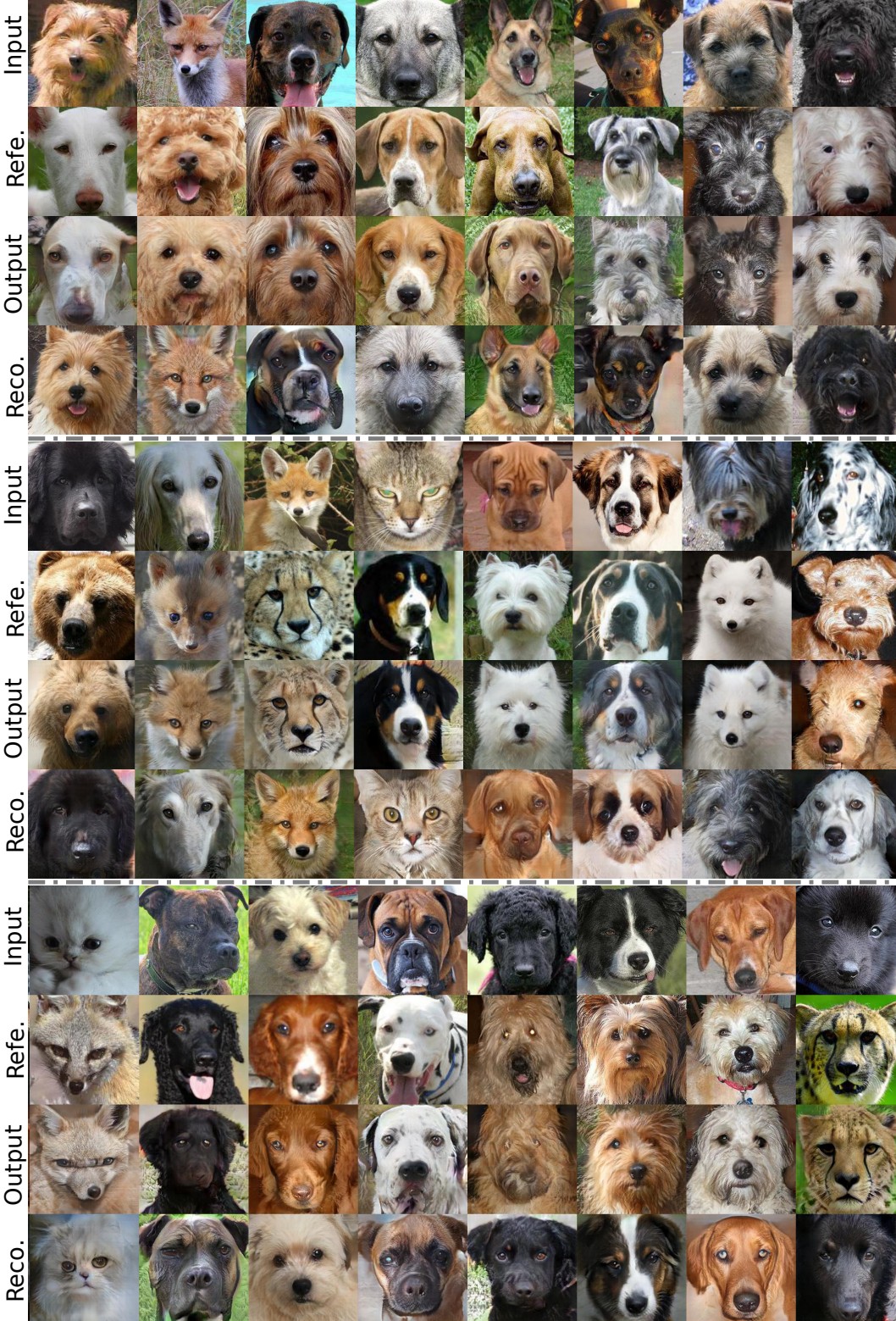

Figure 17: Qualitative results with reference image for image-to-image synthesis task on *Animal* dataset .
Refe.: reference image, Reco.: reconstructed image.

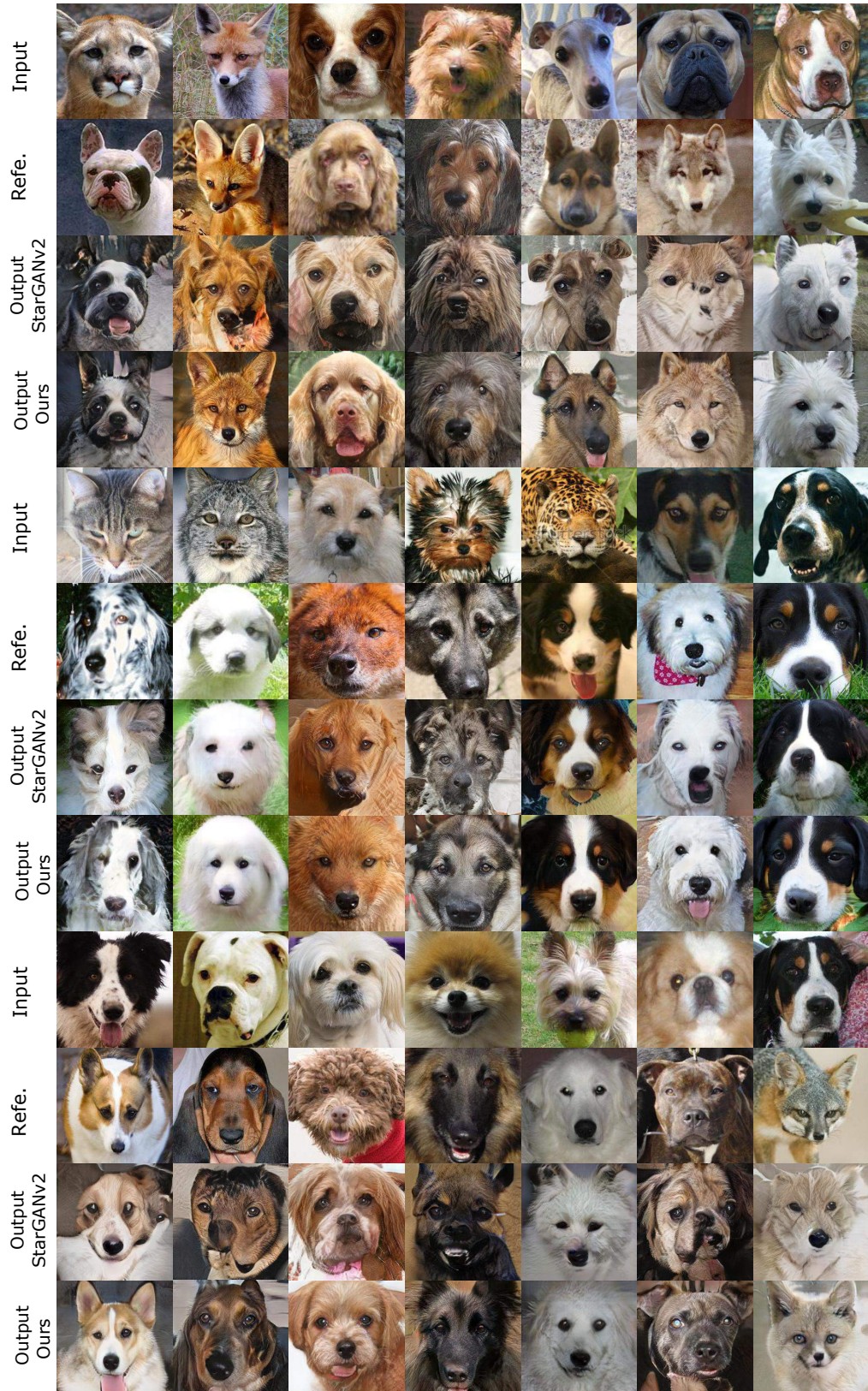

Figure 18: Qualitative comparison on *Animal faces* datasets.

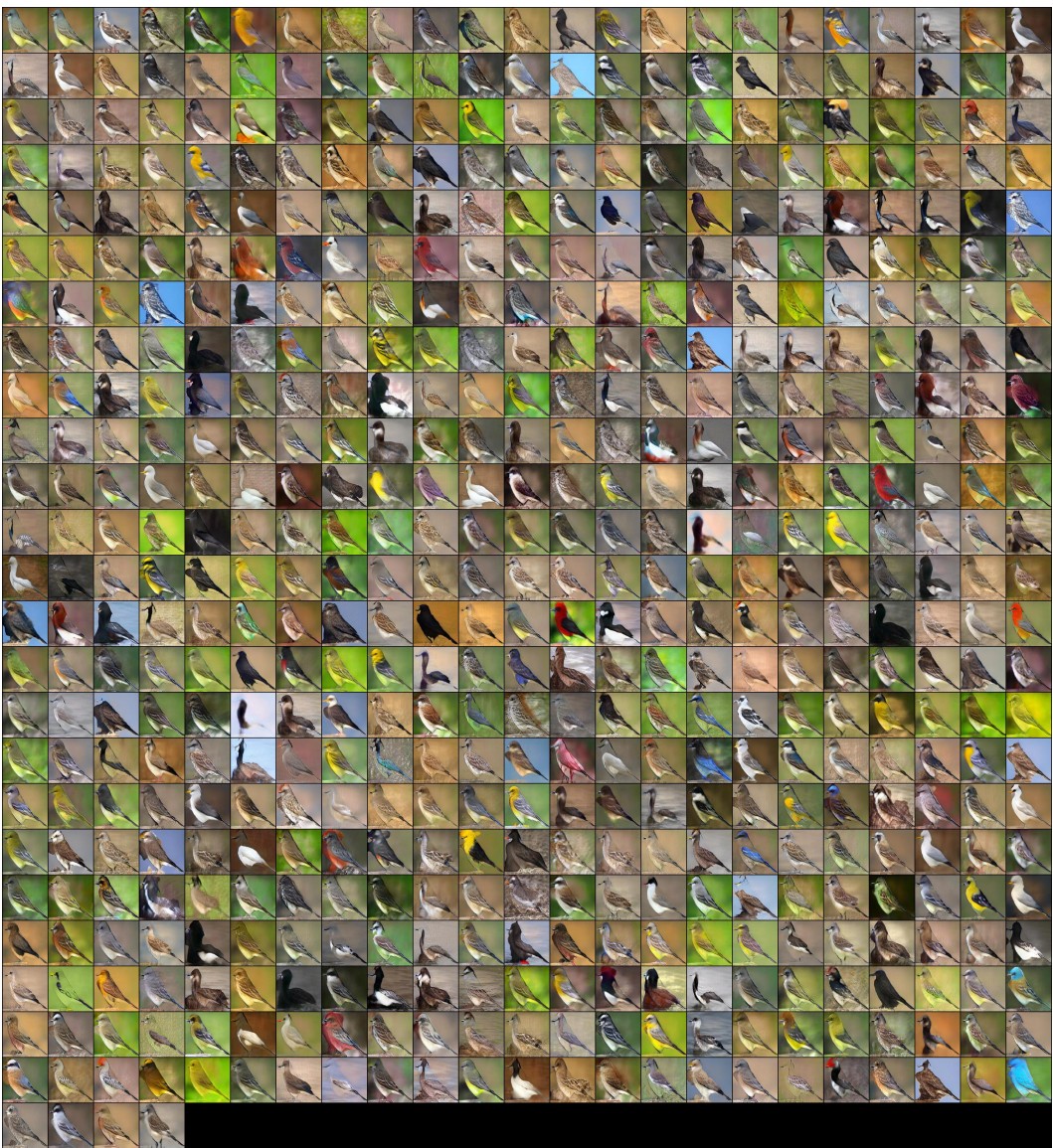

Figure 19: Qualitative results on image-to-image synthesis task. We translate the input image (top left) into all 555 categories. Please zoom-in for details.

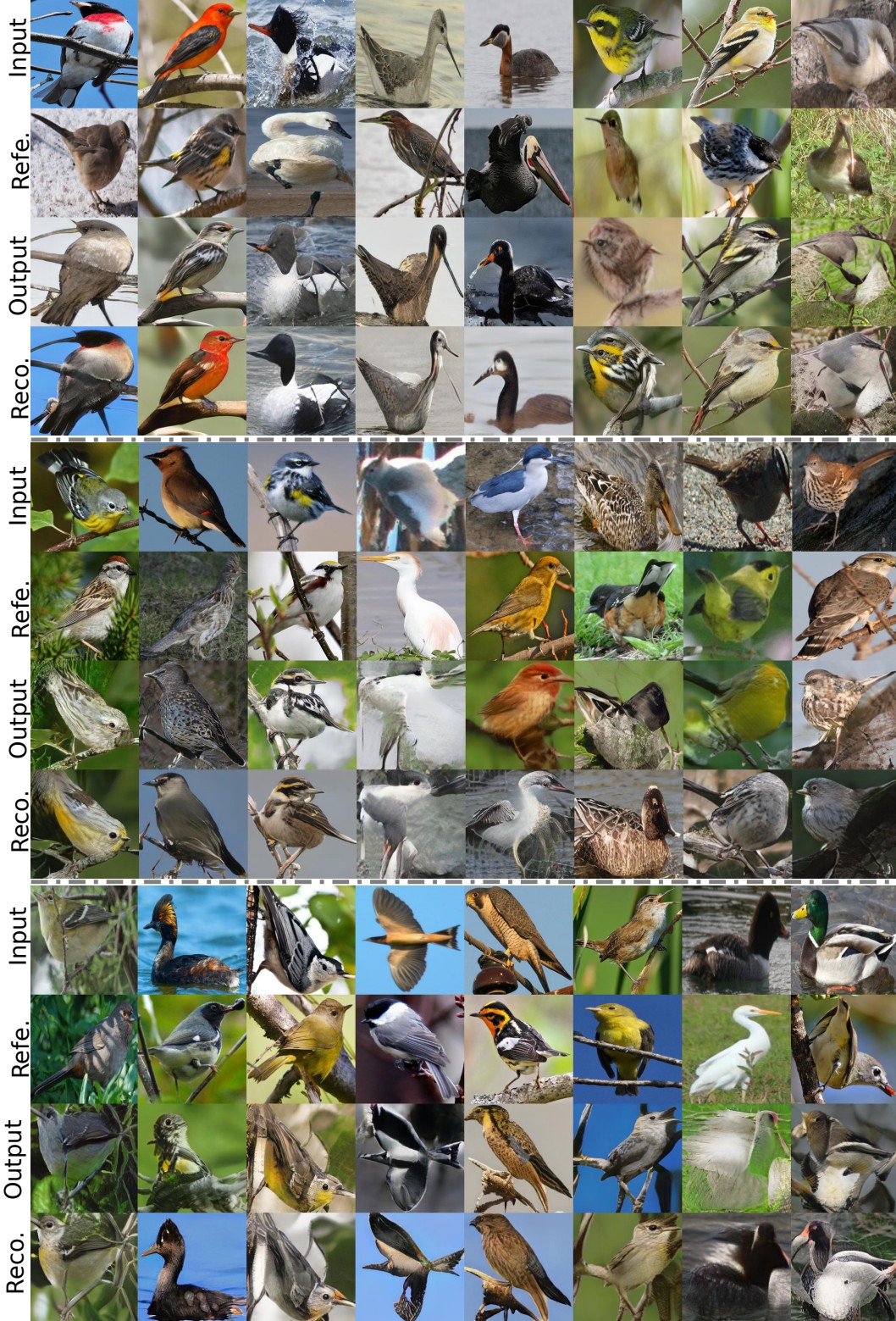

Figure 20: Qualitative results with reference image for image-to-image synthesis task on *Birds* dataset . Refe.: reference image, Reco.: reconstructed image

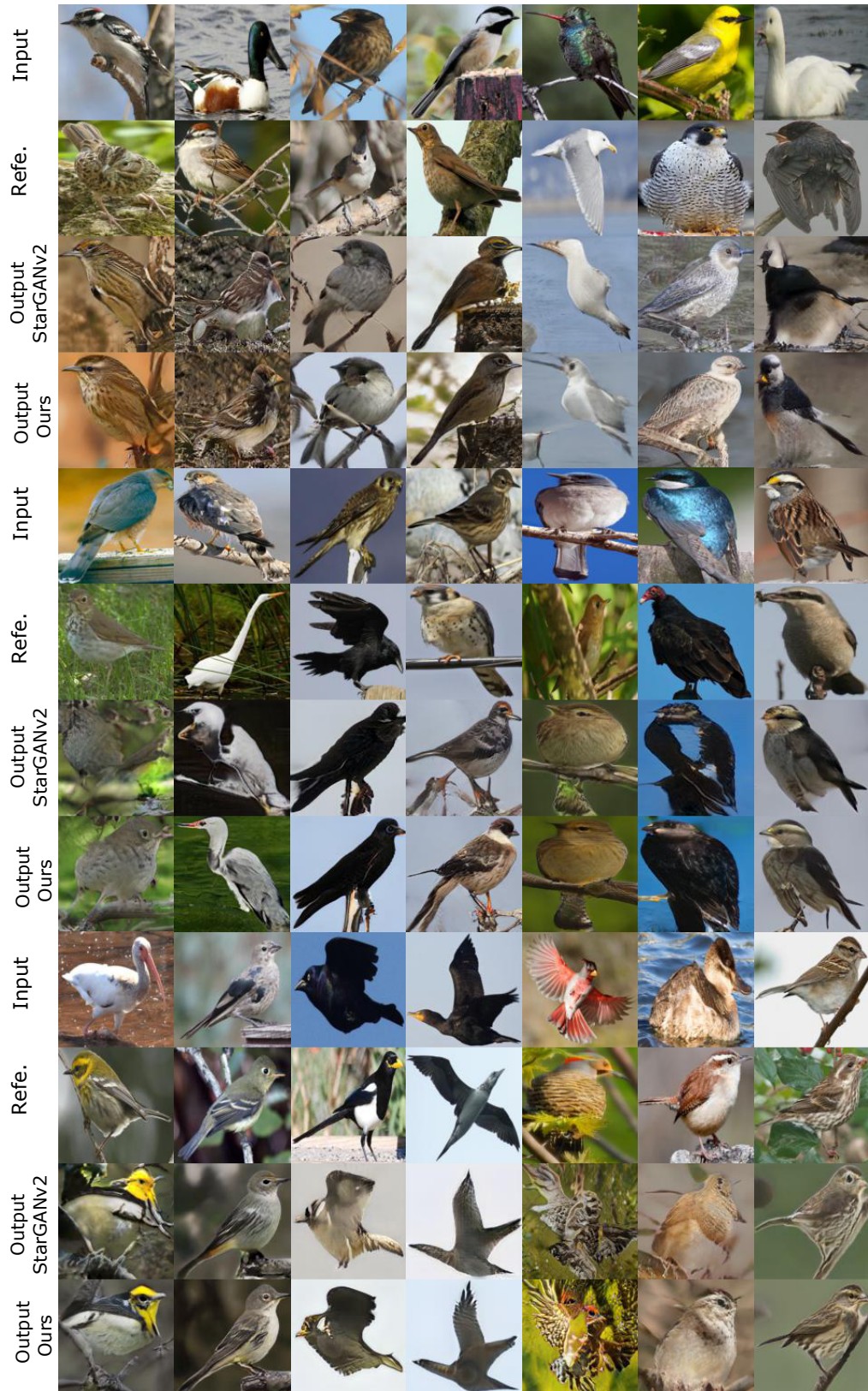

Figure 21: Qualitative comparison on *Birds* datasets.

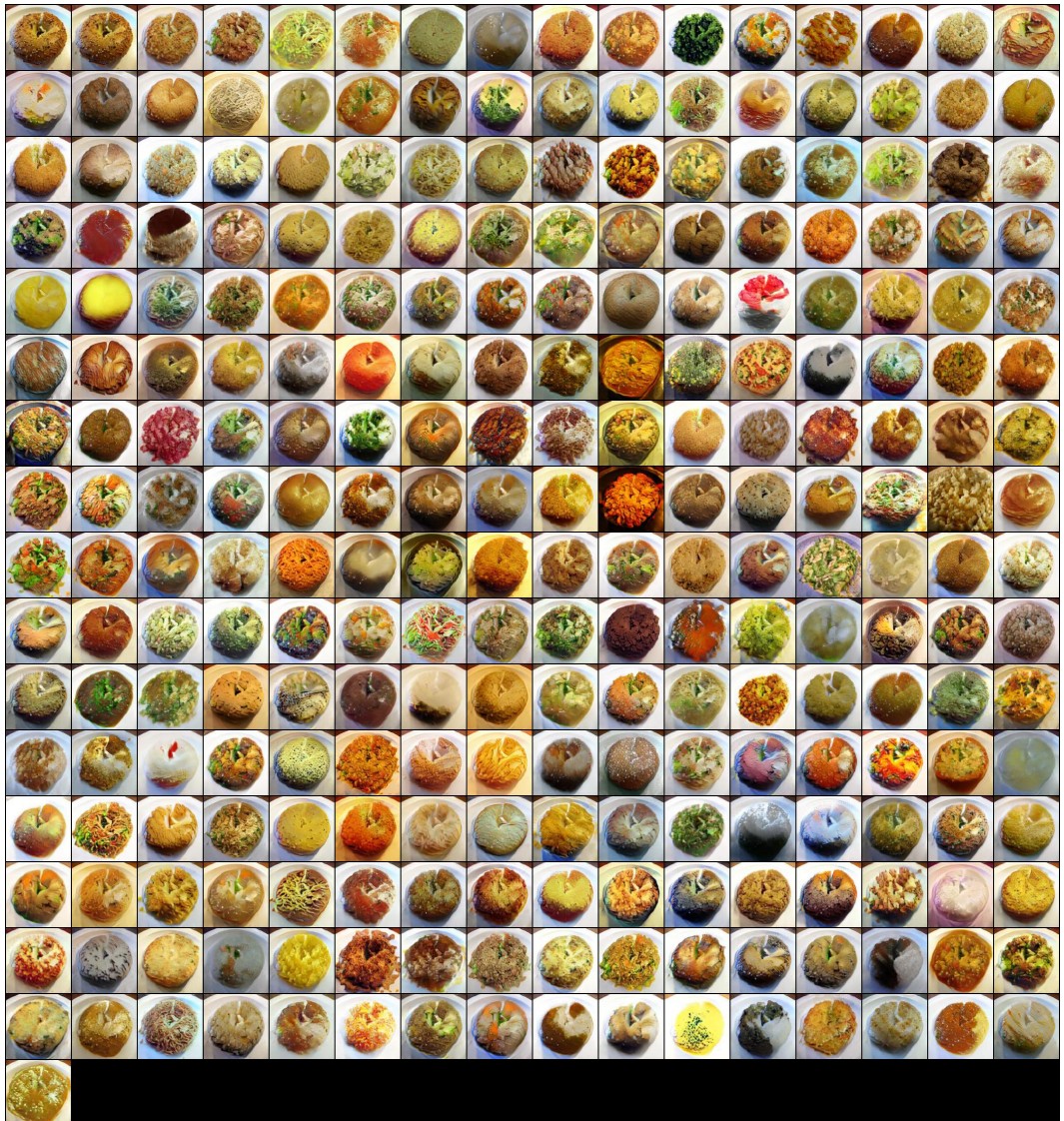

Figure 22: Qualitative results on image-to-image synthesis task. We translate the input image (top left) into all 256 categories. Please zoom-in for details.

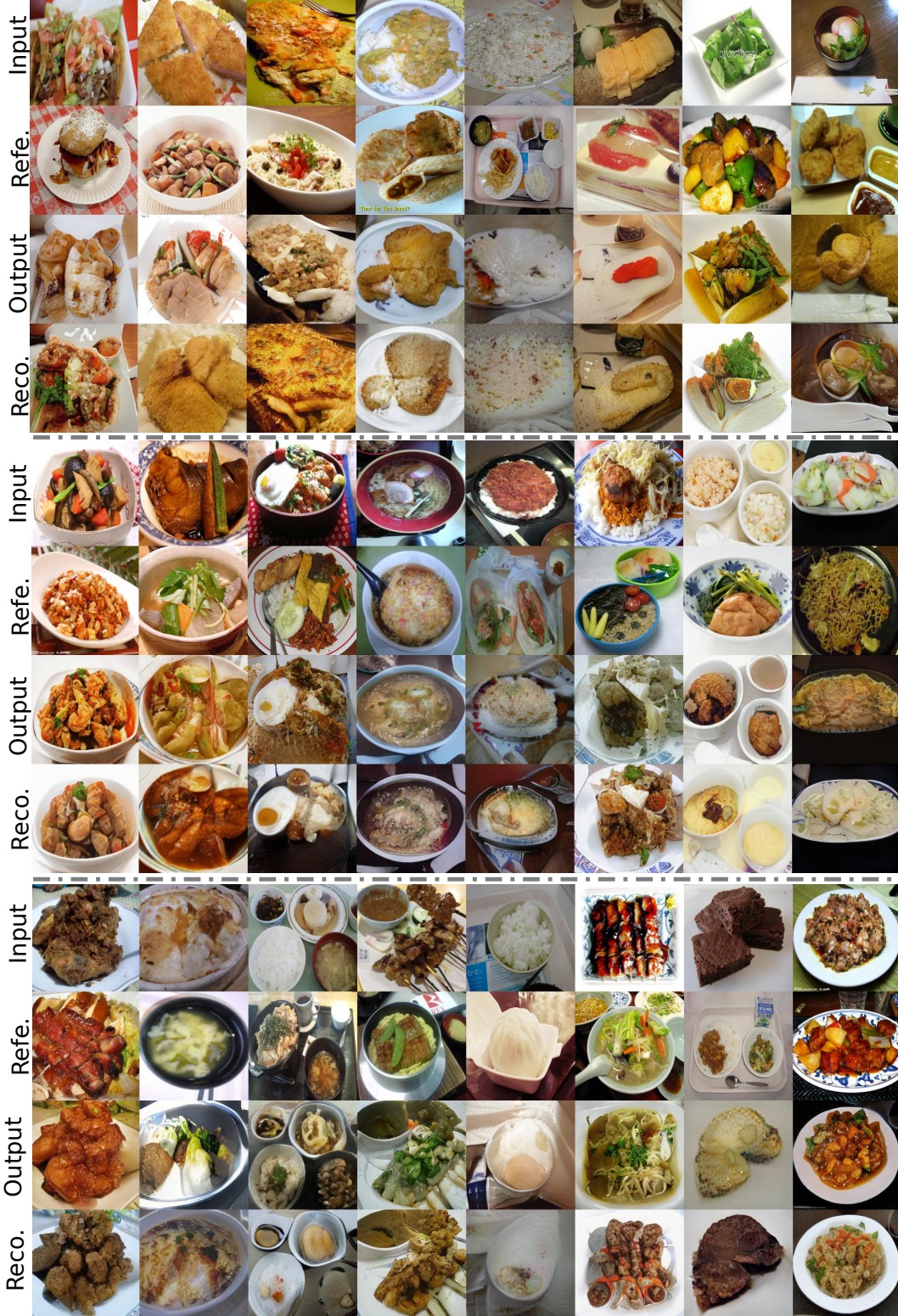

Figure 23: Qualitative results with reference image for image-to-image synthesis task on *Foods* dataset . Refe.: reference image, Reco.: reconstructed image

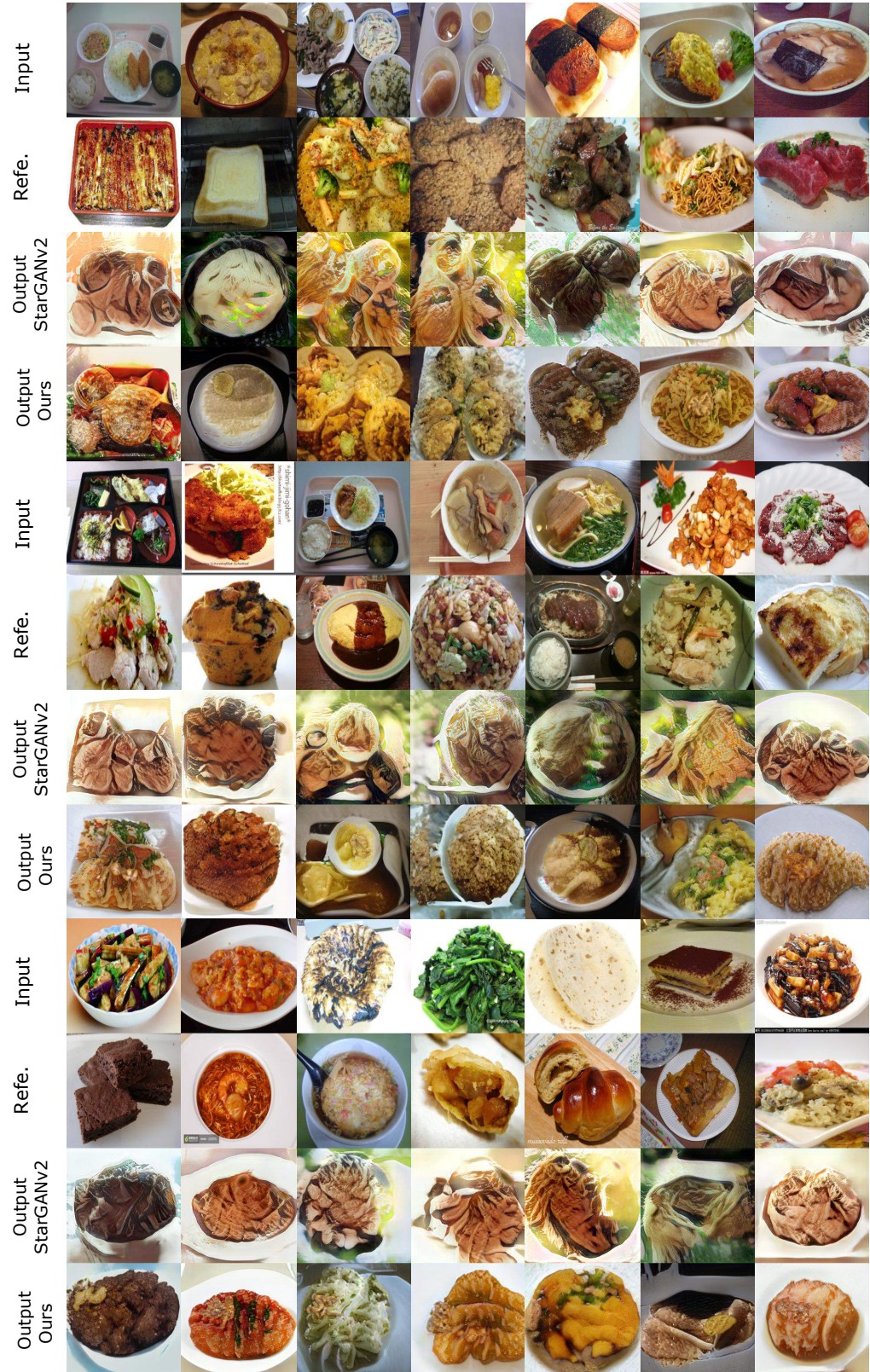

Figure 24: Qualitative comparison on *Foods* datasets.

