# OpenReview forum: "Distilling GANs with Style-Mixed Triplets for X2I Translation with Limited Data"
_ICLR.cc/2022/Conference — ICLR 2022 Poster_

### Official Review · Reviewer_BBix · 2021-11-02

**Correctness:** 3
**Technical Novelty And Significance:** 2
**Empirical Novelty And Significance:** 2
**Recommendation:** 5
**Confidence:** 4

**Main Review:**

-------------------
Strengths
-------------------
- Authors evaluated their approach on multiple datasets and multiple tasks. Demonstrated results prove that qualitatively the model is comparable to modern state-of-the-art such as TransferI2I.
-------------------
Weaknesses
-------------------
- The proposed approach for transfer learning is obviously inspired by TransferI2I. A proper definition of main contributions to the TransferI2I model for the image-to-image translation problem will make the paper more sound.
- The provided comparison with the baselines is limited and doesn't present modern image-to-image translation models, e.g. [a,b].
[a] Schönfeld, Edgar, et al. "You only need adversarial supervision for semantic image synthesis." International Conference on Learning Representations. 2020.
[b] Zhou, Xingran, et al. "CoCosNet v2: Full-Resolution Correspondence Learning for Image Translation." Proceedings of the IEEE/CVF Conference on Computer Vision and Pattern Recognition. 2021.
- The paper is hard to follow, and contains many vague sentences, e,g. Section 3.3, page 5:
However, when naively applying this for knowledge distillation the conditional information can still be ignored
Why conditional information can be ignored?
- The semantic diversity loss is declared to be novel in the abstract. Yet, equation (7) demonstrates that it is a small modification of a loss from paper  by (Yang et al., 2019). The modification is described as 'solely change the conditional terms'. A more detailed explanation why conditional terms must be changed, will significantly improve the description of the proposed semantic diversity loss.



**Summary Of The Paper:**

The paper presents a framework for knowledge distillation in image-to-image translation and arbitrary domain-to-image (e.g., text-to-image) translation. Authors claim following contributions: (1) a unified transfer learning method that leverages only synthetic images, (2) a semantic diversity loss, (3) a style-mixing triplets representing a tuple that includes an input image, a reference image, and a target image.
Authors provide results for image-to-image translation tasks and for text-to-image translation tasks. A limited comparison with several modern models is presented. Authors claim that the proposed transfer learning method outperforms modern state-of-the-art.

**Summary Of The Review:**

The paper would benefit from the following improvements:
- Clear definition of main technical contribution in the Introduction
- Evaluation of the semantic-to-image task in terms of mIoU (i.e. comparison of the input segmentation with the results of a segmentation model on a synthesised image)
- Extended explanation of the semantic diversity loss
- Extended evaluation of the semantic-to-image task with modern models such as [a,b]

---

> ### Author Response · Authors · 2021-11-22
> **Response to Reviewer BBix**
>
> **The proposed approach for transfer learning is obviously inspired by TransferI2I. A proper definition of main contributions to the TransferI2I model for the image-to-image translation problem will make the paper more sound.**
>
> TransferI2I suffers from three problems: 1) It  is a GAN architecture-specific approach, i.e. requiring the GAN architecture within the I2I system to be exactly the same as that of the pretrained GAN. This limits the transfer to current state-of-the-art I2I systems for which no similar GAN architecture exists ( for example StarGANv2 for I2I). 2) It fails to perform reference-guided I2I synthesis, while we propose the style-mixed triplets to achieve this goal.  3) It fails to transfer knowledge  from unconditional GAN to multi-class conditional GAN, while in this paper we have addressed this, e.g. transferring   knowledge from StyleGAN (unconditional GAN) to StarGANv2 (multi-class conditional GAN).
>
>
> **The provided comparison with the baselines is limited and doesn't present modern image-to-image translation models, e.g. [a,b].**
>
>   As suggested by the reviewer, we conduct comparison with the two baselines, the results are as follows. Comparing the new baselines, we still achieve  the best score. Note our method is to transfer knowledge from the StyleGAN to the SPADE based architecture. In fact, we are able to generalize to any  X2I system.  I believe the results could be further improved with our method by  transferring knowledge from StyleGAN to OASIS or CoCosNetv2.   I have added the discussion into section 4.2
>
> |            | FID   | KID  | IS   | mIoU  |
> | ---------- | ----- | ---- | ---- | ----- |
> | SPADE      | 43.91 | 3.26 | 2.53 | 45.3% |
> | OASIS      | 37.60 | 1.74 | 2.75 | 48.6% |
> | CoCosNetv2 | 36.31 | 1.69 | 2.71 | 46.3% |
> | Ours (SPADE based architecture)      | $\textbf{32.98}$ | $\textbf{1.42} $| $\textbf{2.80}$ | $\textbf{50.4}$% |
>
> **The paper is hard to follow, and contains many vague sentences, e,g. Section 3.3, page 5: However, when naively applying this for knowledge distillation the conditional information can still be ignored Why conditional information can be ignore**
>
> We have rewritten this sentence at the start of section 3.3.
>
> In this paper, for X2I translation the training is composed of two steps: data-free distillation from the prerained StyleGAN to X2I translation model (Figure 2(c)) and finetuning X2I translation model using the target data and label.  In the first stage we mention ‘ naively applying this for knowledge distillation the conditional information can still be ignored’.  In this stage, we aim to transfer knowledge without any real data. Thus we explore how to use the input of the conditional branch to influence the output without having access to the target data.
>
> One method is to take as the input  the output of the teacher, and extract the feature embedding, which is taken as the input of the conditional branch of the student generator. We show the generated images of this method in Figure 13 of Appendix C. The visual results show that varying the conditional information fails to significantly influence the output (suggesting that the network ignores the condition). As reported in Table2, this method achieve 90.76 FID on audio2image task, which is worse than our method (70.88 FID).  However, when combined with the semantic diversity loss, the network does use the conditional branch, and our results show that this leads to more efficient knowledge transfer.
>
> Why is the conditional information ignored? In the first stage, to perform the knowledge distillation we expect both the teacher generator and the student generator to have the same output. Thus we align the hierarchical feature outputs and the final image of both the teacher generator and the student generator. The network can solve this task without considering the conditional input (and therefore chooses to ignore it).

---

> > ### Author Response · Authors · 2021-11-22
> > **Response to Reviewer BBix**
> >
> > **The semantic diversity loss is declared to be novel in the abstract. Yet, equation (7) demonstrates that it is a small modification of a loss from paper by (Yang et al., 2019). The modification is described as 'solely change the conditional terms'. A more detailed explanation why conditional terms must be changed, will significantly improve the description of the proposed semantic diversity loss.**
> >
> > We have improved the explanation around Eq. 7. We now describe the difference with (Yang et al. 2019) in more detail.
> >
> > On why it is important that the conditional term changes: In this paper, for X2I translation the training is composed of two stages: data-free distillation from the prerained StyleGAN to X2I translation model (Figure 2(c)) and finetuning X2I generation model using the target data and label.
> > During the first stage we do not use any target data, and we do not have access to the conditional that will be used in the target domain. One approach could therefore be to provide a fixed input to the conditional branch during the initialization (i.e., Ours$\dagger$ ). However, then the weights of the conditional branch need to be learned from scratch on the (small) target domain. To also initialize the conditional branch, we propose one method where the conditional branch takes as input the feature embedding, that is computed from the output of the teacher generator (i.e., Ours$\ddagger$ ).  Finally, we also propose the semantic diversity loss based on the proposed style-mixed triplet.  As reported in Table 2, both  Ours$\dagger$ and Ours$\ddagger$  achieve lower performance ((Ours$\dagger$ , Ours$\ddagger$):(88.81 FID , 90.76 FID) on audio2image task  than our method (70.88 FID). This shows the importance of well-initializing the conditional branch of the network.
> >
> > **Clear definition of main technical contribution in the Introduction**
> >
> > In the initial manuscript, we have summarized the proposed techniques. Here we rewrite our contributions as following: 1) We are the first to investigate knowledge transfer for X2I translation. Therefore, we propose a novel, unified transfer learning method, which can be used for varying kinds of conditional image synthesis tasks (Figure 1) which is based on generated images and therefore does not require any real data.  2) The student generator does not need to have the same architecture as the pretrained GAN. Therefore, we can use well-devised specific image synthesis architectures (e.g., SPADE and StarGANv2) by distilling knowledge from the pretrained teacher GAN (e.g., StyleGAN) to the task-specific student GAN. 3) We use the style mixing characteristic of StyleGAN to create  style-mixed triplet data, which are used to transfer the knowledge efficiently to both I2I and X2I translation models. Furthermore, we propose a semantic diversity loss based on the style-mixed triplet, which contributes to learn the semantic information of the output image.
> >
> > **Evaluation of the semantic-to-image task in terms of mIoU (i.e. comparison of the input segmentation with the results of a segmentation model on a synthesised image)**
> >
> > We compute the  mIoU by using the public model (https://github.com/zllrunning/face-parsing.PyTorch), and the computation code of IoU (https://www.kaggle.com/iezepov/fast-iou-scoring-metric-in-pytorch-and-numpy). The results are  in Table 2 (right) of the revised manuscript . We still achieve better performance in metric mIoU. The results have been added into Table 2 (right).

---

### Official Review · Reviewer_76so · 2021-11-03

**Correctness:** 3
**Technical Novelty And Significance:** 2
**Empirical Novelty And Significance:** 2
**Recommendation:** 6
**Confidence:** 3

**Main Review:**

I think this submission is interesting. It is well written and clearly explains the ideas. I really like the figures which are clean and helpful in understanding the ideas of the paper and performance of the method. I am impressed by the variety of experiments, which answered the questions which arose to me when understanding the method. In the following part, I will detail my comments point by point.

PROS
1. The problem that this submission aims to solve, how to extend the application range of transfer learning for images, is important. And the authors make a step forward.
2. The proposed method is simple and easy to implement, which is also painless to understand.
3. The writting is quite good and I can follow it easily.

CONS

1. While the authors present a number of quantitative comparisons with existing methods, I find the corresponding qualitative results are limited. I can only see Fig. 3 in the submission file. Please attach more visual comparison results to highlight your advantages.
2. How to evaluate the results of each method is quite subjective. And there is not a canonical metric for quantitative evaluations. Thus I suggest the authors do a user study here to reduce the unwanted subjective influences.
3. Currently, this project performs the experiments on the Animal faces, Birds and Foods datasets, which are not the commonly used datasets in many I2I methods. However, with the default settings, they have the best performances on their chosen dataset. So the authors might need to do some experiments on those datasets with corresponding metrics for a fairer comparison.
4. There are also not qualitative comparison figures for tasks of audio2image and text2image.
5. In Fig. 3, I cannot see too much visual improvement especially for the bird figures. For example, StarGAN seems to generate better details. On the $4$-th row,  the feather colors of the proposed method are brighter than StarGAN.
6. Also, another issues is that when the target dataset becomes bigger and more complex, the expected advantages become smaller and the developed method even behaves worse than previous methods (e.g. in Tab. 1 for Foods). This makes me hesitate the practical values of this method when working in the real world, which is very complicated.
7. Are there some examples to support the claim about the style mixing characteristics? Or can the authors present some citations to support it?
8. In Sec. 3.2 & Sec. 3.3, why should use the pre-defined discriminator to initialize the encoder and optimize it? Are there some specific motivations to do so? I assume more discussions are needed here to make it more self-contained.
9. I am looking forward to more special designs when trying to remove the constraint that requires the GAN architecture within the X2I system to be exactly the same as that of the pretrained GAN, instead of justing using several layers for dimension mapping.

**Summary Of The Paper:**

In this paper, the authors present a unifield learning method to investigate the knowledge transfer for X2I translation. Compared to existing methods, there are two advantages here: (1) this framework can be used for varying kinds of conditional image synthesis tasks; (2) it relieves the limitation for student generator to be the same as the pre-trained GAN. Specifically, the authors leverage the style mixing characteristics of high-quality GANs and introduce the semantic diversity loss to achieve it. Many experiments are conducted with good qualitative and quantitative results.

**Summary Of The Review:**

I think the paper could be strengthened by further analysis, experiments, and presentations fixing. However, I like the general idea and I am favorable to accept it considering a large number of work they have done and signifance of this problem.

---

> ### Author Response · Authors · 2021-11-22
> **Response to Reviewer 76so**
>
> **Please attach more visual comparison results to highlight your advantages.**
>
> In the initial manuscript, we have attached more visual comparison results  in appendix C Figures 12, 13, 14, 15. To further highlight our advantage, we additionally show visual comparison results in our revised manuscript (Appendix C, Figures 20, 23, 26).
>
> **User study**
>
> We conduct a user study and ask subjects to select results that are\textit{ more realistic given the target label, and have the same pose as the input image}. The experiment is performed on the  Animal faces dataset. We apply pairwise comparisons (forced choice) with 20 users (100 image pairs/user).   Figure 7 of  the Appendix C.1 shows that ours considerably outperforms the other methods. I have added the result in Appendix C.1 Figure 7, and the corresponding description is in Appendix C.1.
>
> **The authors might need to do some experiments on those datasets with corresponding metrics for a fairer comparison.**
>
>  For manyI2I settings, we evaluated our method on Animal faces, Birds and Foods datasets, which are used in FUNIT[1] and TransferI2I. These datasets contain hundreds of categories, and are more challenging. As suggested by the reviewer , we also leverage the AFHQ dataset [2] (500 per class) to demonstrate the effectiveness of the proposed method. As reported in the table below, we also report the best score in AFHQ-500. We have added this result in Appendix B.2.
>
> |           | AFHQ(500/ per class, latent-guided) |       | AFHQ(500 / per class, reference-guided) |       |
> | --------- | ----------------------------------- | ----- | --------------------------------------- | ----- |
> |           | FID                                 | LPIPS | FID                                     | LPIPS |
> | starganv2 |  40.21                              |  0.45 |  41.62                                  |  0.42 |
> | Ours      |  35.36                              |  0.48 |  34.74                                  |  0.45 |
>
> Furthermore, in the initial manuscript (Appendix B.1), we also evaluate our method on cat2dog datasets, which is widely used in image-to-image translation.
>
> [1] Ming-Yu Liu, Xun Huang, Arun Mallya, Tero Karras, TimoAila,  Jaakko Lehtinen,  and Jan Kautz.   Few-shot unsuper-vised  image-to-image  translation.InProceedings  of  theIEEE International Conference on Computer Vision, pages10551–10560, 2019.
>
> [2] Yunjey Choi, Youngjung Uh, Jaejun Yoo, and Jung-Woo Ha.Stargan v2:  Diverse image synthesis for multiple domains.InCVPR, 2020.
>
> **There are also not qualitative comparison figures for tasks of audio2image and text2image.**
>
> In our initial manuscript, we include the qualitative comparison figures in appendix C (Figures 12-13) due to space limitations in the main  paper.
>
> **In Fig. 3, I cannot see too much visual improvement especially for the bird figures. For example, StarGAN seems to generate better details. On the 4-th row, the feather colors of the proposed method are brighter than StarGAN.**
>
> We exhibit more visual comparison in the revisited manuscript (Appendix C.5, Figures 20,23,26), and believe that we obtain a clear advantage compared to the baseline. Furthermore, as reported in Table1, we obtain better metric values (e.g., 103.2 mFID (ours) vs 128.7 mFID (StarGANv2)), which indicates the effectiveness of the proposed method.
>
> **Also, another issues is that when the target dataset becomes bigger and more complex, the expected advantages become smaller and the developed method even behaves worse than previous methods (e.g. in Tab. 1 for Foods). This makes me hesitate the practical values of this method when working in the real world, which is very complicated.**
>
> We agree with the reviewer's comment: the larger the target data the less the advantage of knowledge transfer. When the target data size is huge, the benefit  of transfer learning is relatively small, since training from scratch would already obtain excellent results. The aim of transfer learning is to transfer knowledge from large labelled datasets to data domains with fewer labels. We believe this to be a problem of large practical impact, and common in many real-world applications. Especially since annotation is often labour intensive, time-consuming  and costly. Therefore, transfer learning is typically evaluated on smaller datasets.
>
> **Are there some examples to support the claim about the style mixing characteristics? Or can the authors present some citations to support it?**
>
> The style-mixing characteristics were discussed in the original StyleGAN paper (see Figure 3). To the best of our knowledge, we are the first one to use style-mixed triplets to perform knowledge transfer. Our insight is that the style mixing characteristics correspond  to the conditional image translation, which motivates us to explore the knowledge transfer from StyleGAN to conditional image translation.

---

> > ### Author Response · Authors · 2021-11-22
> > **Response to Reviewer 76so**
> >
> > **In Sec. 3.2 & Sec. 3.3, why should use the pre-defined discriminator to initialize the encoder and optimize it? Are there some specific motivations to do so? I assume more discussions are needed here to make it more self-contained.**
> >
> > In this paper we use the pre-defined discriminator to initialize the encoder, since the discriminator of the StyleGAN is trained on HHFQ with 70k images, which optimizes it to be an effective feature extractor. The similar technique was also explored in SGD (Shocher et al., 2020) and transferI2I.
> >
> > To verify that the previously reported results also hold for our method, we also train both the encoder and the reference encoder from scratch in reference-guid translation on Animal faces dataset (10/per class). we achieve 130 of mFID, which is lower than our method (mFID 96.2) . If we train the encoder from scratch, the training suffers from overfitting with limited training images.  We have added the results to the Appendix B.4.
> >
> > **I am looking forward to more special designs when trying to remove the constraint that requires the GAN architecture within the X2I system to be exactly the same as that of the pretrained GAN, instead of justing using several layers for dimension mapping.**
> >
> > In existing methods (such as SSA-GAN, S2IGAN and SPADE ) for X2I the parameters of the GAN are trained from scratch. In this work, we show a method based on distillation and the usage of style-mixed triplets that allows for the knowledge transfer from GANs to X2I systems. To the best of our knowledge we are the first work to investigate knowledge transfer for these settings. We do believe that future research in this direction could include more specific knowledge transfer designs but consider that to be future research.

---

> > > ### Comment · Reviewer_76so · 2021-11-30
> > > **Response to the authors after rebuttal**
> > >
> > > First, I really appreciate the long response from the authors. After reading all the comments which address most of my concerns, I hold a positive attitude to this submission. Basically, I think this X2I method is interesting and flexible to many applications. And it also solves (at least tries to solve) an important problem. So I vote to accept.

---

### Official Review · Reviewer_EUGu · 2021-11-09

**Correctness:** 3
**Technical Novelty And Significance:** 3
**Empirical Novelty And Significance:** 2
**Recommendation:** 8
**Confidence:** 4

**Main Review:**

(Update: after reading the reviews and the authors' reply, I would recommend the paper to be accepted to the conference. I'm raising the score from 6 to 8.)

Strengths:
1. The proposed method obtains better image quality against prior work in various settings, as suggested by lower FIDs.
2. Strong results on X2I translation tasks are shown, indicating that the distillation method is effective for X2I tasks.
3. The method improves image translation performance when training on fewer labeled samples.

Weaknesses:
1. For clarification purposes, after the second stage of training of I2I models, can the model explicitly control domains/classes in the latent-guided synthesis, just as how starGANv2 is set up?
2. The clarity of writing can be improved. For example, it takes me a while to understand the setup for the entire training procedure. It might be better to mention more clearly that the distilled model is used as a pre-trained model for the downstream task.
3. Would it be possible to extend the ablation study (Figure 4) to more datasets and settings? It would be more convincing to the reader if the trend is similar in various settings.

Additional questions:
1. During training, I am wondering if the style-mixed triplets are pre-computed and stored, or randomly sampled on the fly. If it is the former case, maybe one can check if increasing the number of stored style-mixed triplets can eliminate the need for knowledge distillation.

**Summary Of The Paper:**

The work introduces a flexible method to distill the knowledge of unconditional GANs of images to various image translation tasks, including image-to-image, text-to-image, and audio-to-image translation. While prior work requires using the same architecture of the pre-trained GANs to train the downstream im2im tasks, the proposed method supports distillation on a wide range of X-to-image translation architectures (e.g. starGANv2). The authors have compared their methods with prior work and showed that the proposed method obtained better FIDs.

**Summary Of The Review:**

This work introduces a nice idea that improves image translation methods by distilling the knowledge of unconditional GANs. The setup is more flexible compared to prior work, where different architectures for image translation can be used. While the clarity of the writing can be improved, I think it is good to accept this paper to the conference.

---

> ### Author Response · Authors · 2021-11-22
> **Response to reviewer EUGu**
>
> **For clarification purposes, after the second stage of training of I2I models, can the model explicitly control domains/classes in the latent-guided synthesis, just as how starGANv2 is set up?**
>
> We are able to explicitly control domains/classes as starGANv2 does. For the latent-guided synthesis, starGANv2 introduces class-specific mapping networks (one for each class) to project the class embedding into the shared latent space. Thus starGANv2 uses the class-specific mapping networks to control the domains/classes. In this paper, we follow the same setup as starGANv2.  In the first stage of our method, we only  learn one mapping network, which is duplicated to initialize all the mapping networks (one for each class) of the second stage. Then we use the target dataset (second stage) to train the well-initialized  class-specific mapping networks. We have added this description into the Appendix A.4.
>
> **The clarity of writing can be improved. For example, it takes me a while to understand the setup for the entire training procedure. It might be better to mention more clearly that the distilled model is used as a pre-trained model for the downstream task.**
>
> Thank you for the suggestion. We have improved the description at the start of section 3 e.g. ‘‘problem setting’’ and the first paragraph of the section 4.
>
> **Would it be possible to extend the ablation study (Figure 4) to more datasets and settings? It would be more convincing to the reader if the trend is similar in various settings.**
>
> Following the suggestion of the reviewer, we  further explore a wide variety of configurations for our approach on  different datasets (i.e., birds dataset and foods dataset) and settings (i.e., audio2image and text2image). We are able to get the similar conclusion:  adding either $\textit{L1} $ distance  or the $\textit{adversarial} $ loss improves the downstream task in general compared to the model trained from scratch. Finally combining both losses obtains the best score, indicating that the proposed method largely improves  image generation performance. We have added the results in Figure 4, Table 2 and section 4.2.
>
> **During training, I am wondering if the style-mixed triplets are pre-computed and stored, or randomly sampled on the fly. If it is the former case, maybe one can check if increasing the number of stored style-mixed triplets can eliminate the need for knowledge distillation.**
>
> We generate style-mixed triplets online based on the images in the minibatch only instead of storing them. Please check the attached code about the details of how we generate style-mixed triplets. We have added a footnote in section  3.1.

---

> > ### Comment · Reviewer_EUGu · 2021-11-29
> > **Response to the authors**
> >
> > I have read the reviews and the author's replies, and I think the author addressed most of the points raised by the reviewers. They clarified the writing in the distillation process, improved baseline comparisons, qualitative comparisons, and ablation studies. I think it is interesting to have a distillation method that is flexible to apply to different X2I architectures, and used it as a pertaining method for the downstream X2I task. Not only it is less restrictive than the prior work, but it also improves the performance. I would still recommend this paper to be accepted, and I'll be raising the score to an 8.

---

### Author Response · Authors · 2021-11-22
**Response to Reviewers**

We thank the anonymous Reviewers for their comments and constructive criticism of our manuscript.  On the basis of reviewer comments we have performed a minor revision of our manuscript. Note that figure, table and bibliographic references have changed and we use those of the new manuscript. In the revised manuscript we have highlighted major changes in the text in red. Figures and tables with captions in red are new in the revised manuscript.

---

### Decision · Program_Chairs · 2022-01-20

**Decision:**

Accept (Poster)

**Comment:**

This paper studies the problem of distilling the knowledge present in different GAN-based image generation tasks. The paper received mixed reviews. The reviewers had difficulty understanding some details regarding the approach, and requests for ablations and clarifications on existing empirical evaluation. The authors provided a strong thoughtful rebuttal that addressed many of those concerns. The paper was discussed and two reviewers updated their reviews in the post-rebuttal phase. Reviewers generally agree that the paper should be accepted but still have concerns regarding contribution and writing. AC agrees with the reviewers and suggests acceptance. However, the authors are urged to look at reviewers' feedback and incorporate their comments in the camera-ready.